# MiSS: Revisiting the Trade-off in LoRA with an Efficient Shard-Sharing Structure

**Jiale Kang**[*]
Yuanshi Inc

**Qingyu Yin**
Zhejiang University

## Abstract

Low-Rank Adaptation (LoRA) is a widely adopted technique for parameter-efficient fine-tuning, but its slow convergence has spurred the development of numerous variants. Nevertheless, existing methods often fail to improve performance, memory footprint, and computational efficiency simultaneously. To address this challenge, we revisit the causes of LoRA's slow convergence. Building on these insights, we propose **M**atrix **S**hard **S**haring (MiSS), which updates shards of the original weight matrix using a single shared trainable matrix $D$, initialized to zeros. To simultaneously ensure computational efficiency, low memory footprint, and scalable serving, we introduce MiSS$^e$. Both theoretical analysis and empirical results demonstrate that our method reduces optimization complexity without compromising performance, thereby achieving a more favorable trade-off among performance, memory, and efficiency. Furthermore, we conduct a comprehensive comparative analysis of various PEFT methods, evaluating their memory usage, initialization overhead, and computational efficiency. By mapping the Pareto frontier across these dimensions, we show that MiSS occupies a favorable position, effectively capturing the advantages of prior approaches.

    🐱 https://github.com/Joluck/MiSS
    🤗 https://github.com/huggingface/peft

## 1 Introduction

Fine-tuning Large Language Models (LLMs) (Radford et al., 2019; Raffel et al., 2020; Yin et al., 2024) is a prevalent methodology for adapting these models to specific downstream tasks. However, full fine-tuning of all parameters is computationally prohibitive. Consequently, numerous Parameter-Efficient Fine-Tuning (PEFT) techniques (Xu et al., 2023) have been developed to mitigate the training expenditure associated with these large-scale models. Among such techniques, Low-Rank Adaptation (LoRA) (Hu et al., 2021) has distinguished itself as one of the most prominent PEFT methods. LoRA employs a low-rank approximation for the weight updates, a strategy that offers a markedly reduced number of tunable parameters, notable efficacy when compared to full fine-tuning, and the potential for zero inference overhead. LoRA constructs this low-rank adaptation matrix through an intuitive design, positing that the weight update $\Delta W$ can be approximated by the product of two lower-rank matrices, $BA \approx \Delta W$. Evidently, this specific factorization is not necessarily the optimal low-rank approximation of the original $\Delta W$.

Many improvements to LoRA have been proposed in recent years, which can be broadly categorized into two major streams: (1) *Adaptability* (Ding et al., 2023; Liu et al., 2024; Biderman et al., 2024): This refers to the convergence speed at which the method reaches an optimal or near-optimal state. The approximation must exhibit a representational capacity comparable to that of the original, full $\Delta W$. Extensive experiments have shown that LoRA's convergence is significantly slower compared to full fine-tuning. To address this issue, researchers have proposed several LoRA variants (Hayou et al., 2024; Meng et al., 2024; Wang et al., 2024a). By adopting different initialization strategies to influence the model's training gradients, they have accelerated LoRA's convergence speed. Different initializations of LoRA variants accelerate convergence essentially by increasing the initial gradients during training or aligning them with the full-scale training gradients. However, many of

---

[*]Correspondence to: kangjiale827@gmail.com

Table 1: A variety of LoRA variants are listed, each with its specific update formulation and initialization strategy for the low-rank matrices. The differences between these methods are compared in a clear and intuitive manner. $^e$ denotes efficient form.

| Method | Forward | Initialization |
|---|---|---|
| LoRA | $y = \boldsymbol{W}_0 x + \boldsymbol{B}\boldsymbol{A}x$ | $\boldsymbol{A} \sim N(0, \sigma^2)$ $\boldsymbol{B} \sim 0$ |
| PiSSA | $y = \boldsymbol{W}_0 x + \boldsymbol{B}\boldsymbol{A}x$ | $\boldsymbol{A} = U_{[:,:r]} S_{[:r,:r]}^{1/2}$, $\boldsymbol{B} = S_{[:r,:r]}^{1/2} V_{[:,:r]}^{\top}$ |
| AdaLoRA | $y = \boldsymbol{W}^{(0)} x + \boldsymbol{P}\boldsymbol{\Lambda}\boldsymbol{Q}x$ | $\boldsymbol{\Lambda} \sim 0$, $\boldsymbol{P}, \boldsymbol{Q} \sim N(0, \sigma^2)$ |
| DoRA | $y = \boldsymbol{m}(\, \boldsymbol{W}_0 x + \boldsymbol{B}\boldsymbol{A}x \,/\, \|\boldsymbol{W}_0 + \boldsymbol{B}\boldsymbol{A}\|_c)$ | $\boldsymbol{A} \sim \text{Rect.KaimingUnif}$, $\boldsymbol{B} \sim 0$ |
| ProLoRA | $y = \boldsymbol{W}_0 x + (\boldsymbol{B_u} \oplus_h \dots)(\boldsymbol{A_u} \oplus_v \dots)x$ | $\boldsymbol{A_u} \sim \text{KaimingUnif}$, $\boldsymbol{B_u} \sim 0$ |
| MoS | $y = \boldsymbol{W}_0 x + \boldsymbol{B}^s \boldsymbol{A}^s x$ | $\boldsymbol{A}^{\text{pub/pri}}, \boldsymbol{B}^{\text{pub/pri}} \sim 0$ |
| **MiSS**(Ours) | $y = \boldsymbol{W_0} x + \text{expand}(\boldsymbol{D})x$ | $\boldsymbol{D} \sim 0$ |
| **MiSS**$^e$(Ours) | $y = \boldsymbol{W_0} x + \boldsymbol{D} \sum_{i=1}^{g} \boldsymbol{x}^{(g)}$ | $\boldsymbol{D} \sim 0$ |

these methods overlook issues of computational efficiency and overall training overhead. For example, PiSSA (Meng et al., 2024) requires a lengthy initialization process, while LoRA-GA (Wang et al., 2024b) depends on modifications to the optimizer, resulting in incompatibility with certain optimizers. (2) *Efficiency* (Kopiczko et al., 2024; Wang et al., 2024c; 2025): This encompasses expeditious initialization, modest memory consumption, and minimal computational overhead. Optimizing LoRA from an efficiency perspective can lead to reduced VRAM consumption and an accelerated training process. Although LoRA has demonstrated significant advantages in reducing parameter scale and computational cost, its effectiveness still falls short of fully matching full fine-tuning. To address this gap, researchers have proposed an increasing number of LoRA variants that gradually approach the performance of full fine-tuning. This raises a natural question:

> *Given the inherent challenge for LoRA and its variants to balance performance, memory, and efficiency, how can we achieve an effective trade-off among all three dimensions?*

To strike a balance between performance, memory, and efficiency, we re-examined the key factors affecting LoRA's slow convergence. Through an analysis of $S^2$FT (Yang et al., 2024), LoRA-FA (Zhang et al., 2023), and LoRA+ (Hayou et al., 2024), we identified a critical phenomenon:

> *During the LoRA fine-tuning process, both matrices $B$ and $A$ need to be updated simultaneously, which increases the complexity of optimization and ultimately leads to slower convergence.*

LoRA+ alleviates this issue by modifying the initial gradients, allowing the fine-tuning process to approximate full fine-tuning better. In contrast, $S^2$FT fixes one matrix as an orthogonal matrix, reducing the degrees of freedom in parameter updates and lowering optimization complexity, thereby enabling faster alignment with the optimal update direction. Inspired by these insights, we hypothesize that training only a single matrix could simplify optimization without sacrificing expressive capacity. We therefore propose **M**atrix **S**hard **S**haring (MiSS), a method that updates a shard of the original weight matrix using a single, shared trainable matrix $\boldsymbol{D}$, initialized to zero. Thus, our approach maintains the low-rank property of the matrices while offering a more efficient alternative to $\boldsymbol{B}\boldsymbol{A}$ updates in terms of computation.

**Gradient Norm Analysis.** We analyze the initial gradient norm to verify our preliminary conclusions. In the experimental sections of the PiSSA, $S^2$FT, and LoRA-GA papers, we observed that LoRA exhibits a very small initial gradient norm compared to full fine-tuning, which shows a much larger one. Notably, all these improved methods share a common characteristic: their initial gradient norms are significantly larger than LoRA, and their early-stage convergence speed is comparable to that of full fine-tuning. Motivated by this, we evaluated the initial gradient norms of different methods across various models and datasets to examine whether MiSS follows the same pattern as other LoRA variants. The experimental results (Figure 1) confirm that MiSS indeed shares this property, i.e., a larger initial gradient norm and faster early convergence. This also supports the hypothesis that optimizing a single matrix is inherently simpler.

Figure 1: Comparison of initial gradient norms across different training methods and the effect of rank. Results are shown for LLaMA2-7B and Qwen3-4B on the Math and Code datasets.

**Efficient Implementation**    To achieve better computational efficiency, we introduce MiSS$^e$, an alternative design that maintains the core principle of parameter sharing while offering improved time and space complexity through input-dimension aggregation. We further conduct extensive experiments (Table 2) to validate its effectiveness.

We first evaluate MiSS on both Natural Language Understanding (NLU) and Generation (NLG) tasks, assessing its performance and scalability. Our results show that MiSS consistently outperforms LoRA and its variants across diverse LLM architectures, establishing new state-of-the-art results on a wide range of metrics. We then analyze the Pareto frontier of the adaptability-efficiency trade-off in PEFT. We argue that an ideal PEFT method should effectively balance these two essential dimensions. To this end, we conduct a series of foundational experiments, including a simulated pre-training and fine-tuning pipeline, computational complexity analysis, and initialization time evaluation. With comprehensive empirical results, we demonstrate that MiSS achieves a favorable balance across three key dimensions **performance, memory, and efficiency**, highlighting its practicality as a general PEFT solution.

Our contributions can be summarized as follows:

1. We propose MiSS, an efficient and adaptable structure with a shard-sharing mechanism, striking an effective balance among three essential properties—performance, memory efficiency, and computational efficiency.

2. Through large-scale experiments across diverse datasets and model architectures, we provide a comprehensive evaluation of multiple PEFT methods. Our empirical results conclusively demonstrate that MiSS achieves a superior balance among these three properties compared to existing alternatives.

## 2    PRELIMINARIES AND RELATED WORKS

**Low-Rank Adaptation (LoRA).**    Parameter-Efficient Fine-Tuning (PEFT) refers to a family of techniques designed to adapt large pre-trained models to downstream tasks while minimizing the number of trainable parameters, thereby reducing computational and memory overhead. Among diverse methods, Low-Rank Adaptation (LoRA) has gained significant prominence. It operates on the principle that the change in weights during model adaptation often possesses a low intrinsic rank. Instead of fine-tuning the entire pre-trained weight matrix $\boldsymbol{W}_0 \in \mathbb{R}^{d \times k}$, LoRA introduces a low-rank decomposition to represent the update. Consider a simple linear projection with input $x \in \mathbb{R}^d$ and output $y \in \mathbb{R}^k$, LoRA adapts the following forward pass:

$$y = (\boldsymbol{W}_0 + \boldsymbol{\Delta W})x \approx \boldsymbol{W}_0 x + \boldsymbol{B}\boldsymbol{A}x, \text{ where } \boldsymbol{B} \in \mathbb{R}^{d \times r}, \ \boldsymbol{A} \in \mathbb{R}^{r \times k}. \tag{1}$$

Here, $\boldsymbol{A}$ and $\boldsymbol{B}$ are low-rank matrices, with the rank $r$ being significantly smaller than the original dimensions *i.e.,* $r \ll \min(d, k)$. During the fine-tuning process, the original weights $\boldsymbol{W}_0$ are kept frozen, and only the parameters within matrices $\boldsymbol{A}$ and $\boldsymbol{B}$ are trained. Specifically, LoRA initializes $\boldsymbol{A}$ with Gaussian noise $A \sim N(0, \sigma^2)$ with small $\sigma$ and $B$ with zeros, ensuring that $\boldsymbol{B}\boldsymbol{A} = 0$ at the start, preserving the pre-trained model's output.

**Improvements of LoRA.**    LoRA is the low rank adaptation towards full-param finetuning, and intuitively it downperforms than it. Several works propose diverse methods towards a better convergence and adaptability of LoRA. One compelling venue is to change the form of LoRA. PiSSA (Meng et al., 2024) optimizes the compact parameter space by representing the matrices in the model as the product of two trainable matrices, augmented with a residual matrix for error

correction. Using Singular Value Decomposition (SVD), OLoRA (Büyükakyüz, 2024) leverages QR decomposition to initialize the adaptation matrices during the fine-tuning process, ensuring that these matrices are orthogonal. This orthogonal initialization helps maintain the stability of the parameter space during optimization. LoRA-GA and PiSSA are similar in form, but they differ in that LoRA-GA initializes $A$ and $B$ by computing the initial gradient, thereby closely approximating full fine-tuning. LoRA+ extended this method by introducing independent learning rates for matrices $A$ and $B$ with a fixed ratio, improving the method's efficiency. DoRA (Liu et al., 2024) decomposes the weight matrix into two parts: magnitude and direction, which are optimized separately. This approach allows for more precise control over the learning rate, making LoRA updates closer to the effect of full fine-tuning. The improvements brought by these LoRA variants validate that the updates to the weights exhibit a low intrinsic rank during adaptation and hold greater potential. However, they also introduce more complex initialization steps and increase preprocessing time.

# 3 NO FREE LUNCH: BALANCING BETWEEN ADAPTABILITY AND EFFICIENCY

This section elucidates the fundamental trade-off inherent in LoRA-style PEFT techniques: the delicate balance between their *adaptability* and *efficiency*. Adaptability, in this context, refers to the capacity of a given method to emulate the performance benchmarks set by full-parameter fine-tuning. Conversely, efficiency encompasses the method's judicious use of computational resources, specifically time and memory. We utilize highly artificial controlled dataset and model with a relatively small parameter count to make the verification transparently and easy for replication.

We considered diverse methods [1]: (1) Full-parameter finetuning (Lv et al., 2024). (2) LoRA (Hu et al., 2021). (3) Alternatives to LoRA *w/* different architectures, including: PiSSA (Meng et al., 2024), VeRA (Kopiczko et al., 2024), DoRA (Liu et al., 2024) and MoRA (Jiang et al., 2024). (4) Efficent LoRA Design that keeps the LoRA $BA$ structure: PROLORA (Wang et al., 2024c), MoS (Wang et al., 2025). (1) An overview of their forward form, initialization method can be found at Table 1.

## 3.1 EMPIRICALLY BENCHMARKING THE ADAPTABILITY OF LORA VARIANTS

**Experimental Setup.** Parameter-efficient adaptation methods, particularly those leveraging low-rank principles, typically constrain trainable parameters by applying low-rank decompositions either to newly introduced adapter matrices or to the updates of pre-existing model weights. To rigorously evaluate such strategies, we selected a deliberately minimalistic base model: a single-layer MLP designed to process a series of features and yield outputs. This model is initially pre-trained to fit some sinusoidal functions using a constrained set of data points. Following this pre-training, the target function is subtly altered, and an additional dataset sampled from this modified function is employed for training to assess the adaptation performance of various fine-tuning techniques. Comprehensive details regarding the experimental settings are elaborated in Appendix C.

**Results.** Figure 2 illustrates the comparative adaptability of different methods. We utilize the minimum validation loss achieved by each approach as an indicator of its expressive capacity when approximating the performance of full-parameter fine-tuning. The results clearly demonstrate that methods leveraging singular value decomposition (SVD), such as PiSSA, attain a relatively low loss. Conversely, efficiency-focused techniques like MoS exhibit higher losses. A plausible explanation for this discrepancy is that such methods further decompose LoRA matrices into shared components, which may inherently constrain their expressive power. Our method MiSS reaches a relatively advanced performance comparing to other variants.

---

[1]We have not included methods such as LoRA-GA (Wang et al., 2024b) or LoRA+ (Hayou et al., 2024) in our current analysis. While these approaches aim to more closely approximate the performance of full-parameter fine-tuning, we consider MiSS to be largely orthogonal to them. Consequently, the analytical techniques employed in their study may still offer valuable insights for MiSS.

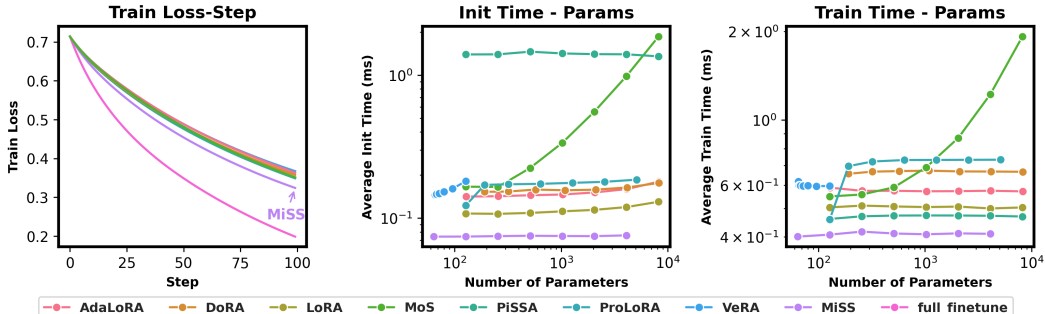

Figure 2: No Free Launch Experiment. **Left.** The training loss curves of all methods. **Middle.** Initialization time *w/* parameters. **Right.** Training time *w/* parameters.

## 3.2 EFFICIENCY ANALYSIS OF LORA VARIANTS

**Metrics.** We evaluate the efficiency of LoRA-like variants from two primary perspectives: (1) *Space and Time Complexity in Training*. Space and time complexity during training are generally considered crucial criteria for evaluating PEFT methods. To benchmark these aspects, we employ the model architecture detailed in Section 3.1. We also test the real cost in our experiment section *i.e.,* Section 5.3. (2) *Initialization*. Initialization time is often overlooked in theoretical complexity analyses. This oversight typically stems from the assumption that common initialization techniques (e.g., Kaiming Initialization) are computationally inexpensive and represent a one-time cost within the entire training pipeline. However, several recent advancements in LoRA and its variants incorporate matrix operations (e.g., Singular Value Decomposition - SVD) that are not inherently hardware-friendly and can pose challenges for efficient optimization and computation. Consequently, we explicitly include initialization time as a distinct evaluation metric in our experimental framework. We then progressively scale the trainable parameter count of various approaches to meticulously measure their respective time and space costs.

**Results.** The efficacy (See Figure 2) of MiSS is evident: its strategic combination of parameter sharing and an efficient computational design culminates in rapid, scalable performance across both initialization and training stages. In contrast, while techniques like PiSSA demonstrate commendable adaptability, as shown in prior experiments, their reliance on computationally intensive Singular Value Decomposition for initialization significantly hampers their overall speed. Other approaches, such as VeRA and AdaLoRA, offer efficient initialization and computation; however, as previously discussed, they often achieve this at the cost of comparatively reduced adaptability.

## 4 MISS: SHARD SHARING FOR THE PERFORMANCE AND EFFICIENCY TRADEOFF

### 4.1 METHOD OVERVIEW

In traditional low-rank adaptation methods *e.g.,* LoRA, the weight update $\Delta W$ is approximated as a low-rank matrix, e.g., $\Delta W = BA$, where $A \in \mathbb{R}^{r \times k}$, $B \in \mathbb{R}^{d \times r}$, and the rank $r \ll \min(d, k)$. This approach achieves efficiency by limiting the number of parameters. However, we observe that a repeating matrix—where a small matrix is replicated to form a larger one—can also be viewed as a low-rank structure. For instance, if a matrix's rows or shards are constructed by repeating a limited set of independent elements, its effective rank is often much smaller than its full dimensions.

Based on this insight, we propose MiSS, which defines the weight update $\Delta W$ as a large matrix generated from a small trainable matrix $D$ through an expansion operation. The updating of $W$ and the forward pass can be expressed as:

$$W = W_0 + \Delta W = W_0 + \text{expand}(D), \quad y = W_0 x + \text{expand}(D)x. \tag{2}$$

Here, $x \in \mathbb{R}^{b \times l \times k}$, $y \in \mathbb{R}^{b \times l \times d}$, $W_0 \in \mathbb{R}^{d \times k}$ is the pre-trained weight matrix, $D \in \mathbb{R}^{r_1 \times r_2}$ is a small trainable matrix with $(r_1, r_2) \ll \min(d, k)$, and $\text{expand}(D)$ is a function that extends $D$ to $\mathbb{R}^{d \times k}$. This structure inherently exhibits low-rank properties. Since the rows within each shard are

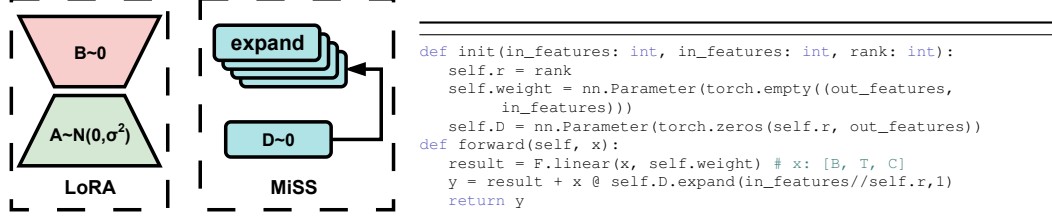

Figure 3: **Left.** Structural diagram of $\Delta W$ in LoRA and MiSS. **Right.** PyTorch-style pseudocode illustrating the implementation of MiSS.

identical, the rank of $\mathrm{expand}(\boldsymbol{D})$ is at most $N$. When $N \ll d$, $\Delta W$ is a low-rank matrix, reducing the parameter count from $d \times k$ to $N \times k$.

Regarding the expansion method, we partition the output dimension $d$ of $\boldsymbol{W_0}$ into $N$ shards of sizes $\{s_1, s_2, \ldots, s_N\}$, where $\sum_{i=1}^{N} s_i = d$. Let $\boldsymbol{D} \in \mathbb{R}^{N \times k}$, where $N$ is the number of shards. For each shard $i$, its update is determined by the $i$-th row of $\boldsymbol{D}$, denoted $\boldsymbol{D_i} \in \mathbb{R}^{1 \times k}$, repeated $s_i$ times to form the shard's update matrix. Formally:

$$(\mathrm{expand}(\boldsymbol{D}))^{\mathsf{T}} = [(\mathbf{1}_{s_1}\boldsymbol{D_1})^{\mathsf{T}} \ (\mathbf{1}_{s_2}\boldsymbol{D_2})^{\mathsf{T}} \ \ldots \ (\mathbf{1}_{s_N}\boldsymbol{D_N})^{\mathsf{T}}] \tag{3}$$

Here, $\mathbf{1}_{s_i} \in \mathbb{R}^{s_i \times 1}$ is an all-ones vector, and $\mathbf{1}_{s_i}\boldsymbol{D_i}$ denotes $\boldsymbol{D_i}$ repeated $s_i$ times vertically. The shards are vertically concatenated to match the dimensions of $\boldsymbol{W_0}$.

## 4.2 Efficient Implementation of MiSS

The above formulation is effective in the initialization process, as it only needs to initialize a small $\boldsymbol{D}$. However, directly computing $\mathrm{expand}(\boldsymbol{D})x$ has a time complexity of $O(bldk)$ and memory complexity of $O(dk)$, which can be computationally intensive. It is obvious that MiSS can be transformed into an efficient form that leverages the block structure of the input to avoid explicitly forming the large matrix, by redefining $\boldsymbol{D} \in \mathbb{R}^{d \times r}$, where $r$ is a tunable rank parameter. Instead of partitioning the output dimension $d$, we divide the input dimension $k$ into $r$ blocks, each of size $g = \lfloor k/r \rfloor$ (for simplicity, assume $k$ is divisible by $r$). For an input $\boldsymbol{x} \in \mathbb{R}^{b \times l \times k}$, partition it along the $k$-dimension, and sum each block along the $k$-dimension:

$$\boldsymbol{x^{(i)}} = \boldsymbol{x}_{[:,:,(i-1)*r:i*r]} \in \mathbb{R}^{b \times l \times r} \tag{4}$$

$$\boldsymbol{x} = [\boldsymbol{x^{(1)}}, \boldsymbol{x^{(2)}}, \ldots, \boldsymbol{x^{(g)}}] \tag{5}$$

$$\boldsymbol{S} = \sum_{i=1}^{g} \boldsymbol{x^{(g)}} \in \mathbb{R}^{b \times l \times r} \tag{6}$$

This enjoys the following updating term and forward pass:

$$\Delta \boldsymbol{W} \boldsymbol{x} = \boldsymbol{D} \boldsymbol{S}, \ \ \boldsymbol{y} = \boldsymbol{W_0} \boldsymbol{x} + \boldsymbol{D} \boldsymbol{S}, \ \text{where } \boldsymbol{D} \in \mathbb{R}^{d \times r}. \tag{7}$$

Here $\boldsymbol{S} \in \mathbb{R}^{b \times l \times r}$, and $\boldsymbol{D} \boldsymbol{S} \in \mathbb{R}^{b \times l \times d}$, matching the dimensions of $W_0 x$.

This efficient form implicitly defines $\mathrm{expand}(\boldsymbol{D})$, such that $\mathrm{expand}(\boldsymbol{D})x = \boldsymbol{D} \boldsymbol{S}$. Specifically, $\mathrm{expand}(\boldsymbol{D}) \in \mathbb{R}^{d \times k}$ has rows corresponding to rows of $\boldsymbol{D}$, repeated across blocks in the $k$-dimension. *E.g.*, if $k = 6$, $r = 3$, and $g = 2$, the $i$-th row of $\mathrm{expand}(\boldsymbol{D})$ takes values $\boldsymbol{D}_{j,i}$ in block $j = \lceil j'/g \rceil$, where $j'$ is the column index. This structure avoids storing the $d \times k$ matrix explicitly, requiring only $\boldsymbol{D} \in \mathbb{R}^{d \times r}$, significantly reducing memory usage.

The efficient implementation of MiSS relies on an innovative input aggregation mechanism, namely blockwise input summation. We highlight its advantages through the following steps: (1) *Input Partitioning and Aggregation*: The aggregation exploits local redundancy in the input, preserving critical information while reducing the computational dimensionality. (2) *Fast Computation*: The cost of computing the efficient form is significantly lower than the original complexity. (3) *Resource Savings*: Memory usage drops comparing to original form.

### 4.3 SYSTEMATIC ANALYSIS OF MEMORY AND EFFICIENCY FOR LORA AND MISS

This subsection systematically compares LoRA variants against MiSS, dissecting their intrinsic differences in memory consumption (governed by parameter count) and computational efficiency (governed by FLOPs and operator type). Our analysis centers on the core update formulations: $\Delta\mathbf{W}\mathbf{x} = \mathbf{B}\mathbf{A}\mathbf{x}$ for LoRA, versus $\Delta\mathbf{W}\mathbf{x} = \mathbf{D}\mathbf{S}$ for the efficient form of MiSS (MiSS$^e$), where $\mathbf{S}$ denotes the blockwise input aggregation. We denote the LoRA rank as $r_\mathrm{L}$, MiSS rank as $r_\mathrm{M}$, with input dimension $k$ and output dimension $d$.

**Limitations of LoRA Variants: Parameter Reduction $\neq$ Computational Speedup**   As illustrated in Table 2, there exists a fundamental misalignment between parameter efficiency and computational cost in existing PEFT methods. While variants like AdaLoRA, DoRA, and VeRA significantly reduce Trainable Parameters (TPs) through novel initialization or decomposition strategies, they almost universally inherit the sequential matrix multiplication logic $\mathbf{B}(\mathbf{A}\mathbf{x})$. Consequently, their **Space Complexity** and **FLOPs** remain bound by the $O((d+k) \times r)$ lower limit. Furthermore, sophisticated variants such as LoHA introduce additional structural overhead (e.g., the $2r$ factor), causing actual memory occupancy and latency to exceed the original LoRA despite having fewer trainable parameters.

Table 2: Comparison of PEFT Methods. Note that while distinct LoRA variants reduce TPs, they fail to improve Space Complexity and FLOPs due to the unchanged sequential computation, unlike the proposed MiSS.

| Methods | Space Complexity | FLOPs | TPs |
|---|---|---|---|
| FT | $O(d \times k)$ | $O(d \times k)$ | $d \cdot k$ |
| LoRA | $O((d+k) \times r)$ | $O((d+k) \times r)$ | $(d+k) \cdot r$ |
| LoRA-FA | $O((d+k) \times r)$ | $O((d+k) \times r)$ | $d \cdot r$ |
| AdaLoRA | $O((d+k+r) \times r)$ | $O((d+k+r) \times r)$ | $(d+k) \cdot r + r^2$ |
| LoHA | $O(2r \times (d+k))$ | $O(2r \times (d+k))$ | $2 \cdot (d+k) \cdot r$ |
| VeRA | $O((d+k)r + r + d)$ | $O((d+k)r + r + d)$ | $d + r$ |
| **MiSS$^e$** | $\boldsymbol{O(d \times r)}$ | $\boldsymbol{O(k + d \times r)}$ | $\boldsymbol{d \cdot r}$ |

**Single-Matrix Paradigm and Computational Decomposition**   MiSS fundamentally diverges from the standard LoRA architecture by employing a single low-rank matrix $\mathbf{D} \in \mathbb{R}^{r_1 \times r_2}$, rather than the dual-matrix structure $(\mathbf{A}, \mathbf{B})$. Crucially, we observe that $\mathbf{D}$ in MiSS$^e$ is dimensionally consistent with $\mathbf{B}$ in LoRA, as both correspond to the output dimension $d$ and function as the output operation matrix. This structural alignment allows us to naturally decompose the computation into two distinct stages: *Input Transformation* ($C_{\text{Step 1}}$) and *Output Projection* ($C_{\text{Step 2}}$). This insight isolates the efficiency distinction entirely to $C_{\text{Step 1}}$. While LoRA relies on an expensive matrix multiplication ($\mathbf{A}\mathbf{x}$), MiSS$^e$ utilizes a cost-efficient block summation (sum($\mathbf{x}$)). The comparative analysis is summarized below:

Table 3: Computational Decomposition of MiSS$^e$ vs. LoRA

| Metric | LoRA | MiSS$^e$ |
|---|---|---|
| **Structure** | Dual Matrices $(\mathbf{A}, \mathbf{B})$ | Single Matrix $(\mathbf{D})$ |
| $C_{\text{Step 2}}$ **(Output Projection)** | Matrix Mult. $\mathbf{B}\mathbf{h}$ $(d \times r)$ | Matrix Mult. $\mathbf{D}\mathbf{S}$ $(d \times r)$ |
| $C_{\text{Step 1}}$ **(Input Transform)** | **Matrix Mult. $\mathbf{A}\mathbf{x}$** $(O(BLkr))$ | **Block Sum** sum($\mathbf{x}$) $(O(BLk))$ |
| Parameter Count $(N)$ | $O(r(k+d))$ | $O(rd)$ |
| Total FLOPs | $O(BL(kr + rd))$ | $O(BL(k + rd))$ |

## 5 EXPERIMENTS

In this section, we conduct a comprehensive set of experiments to validate the effectiveness and generalizability of MiSS across diverse domains. We assess performance on a wide range of tasks, including **language, image, and video benchmarks**. Specifically, we evaluate Natural Language Understanding (NLU) capabilities using a subset of the GLUE dataset, and Natural Language Generation (NLG) capabilities by fine-tuning various large language models (LLMs). We extend our

evaluation to multimodal settings using the VTAB-1K benchmark to demonstrate the robust adaptability of MiSS beyond textual domains. Furthermore, we provide a detailed analysis of the Pareto frontier (Section 5.3) to definitively illustrate MiSS's superior computational efficiency and minimal hardware overhead when compared to existing Parameter-Efficient Fine-Tuning (PEFT) methods.

## 5.1 SUPERIOR PERFORMANCE ACROSS LANGUAGE AND VISION DOMAINS

MiSS demonstrates exceptional versatility, maintaining a commanding lead or highly competitive performance across diverse benchmarks in both the language and vision domains. (Setup B)

**Natural Language Understanding (NLU).** On the GLUE benchmark (Table 4), fine-tuning RoBERTa-base with MiSS showcases notable strength. It achieves an outstanding result on the challenging CoLA dataset (**72.86**), significantly surpassing LoRA and PiSSA. This performance indicates superior data-fitting capabilities and faster convergence on complex linguistic tasks.

Table 4: The results of fine-tuning RoBERTa-base using MiSS and various LoRA variants were compared on a subset of the GLUE benchmark.

| Method | Trainable | MNLI | SST-2 | CoLA | QNLI | MRPC | Avg |
|---|---|---|---|---|---|---|---|
| LoRA | 0.236% | 85.63±0.01 | **94.03±0.02** | 62.40±0.71 | 91.37±0.97 | 87.98±0.23 | 84.28 |
| PiSSA | 0.236% | **85.72±0.40** | 93.64±0.13 | 67.28±0.59 | 91.40±0.54 | 88.11±0.24 | 85.23 |
| MiSS | 0.236% | 85.71±0.32 | 93.60±0.07 | **72.86±3.13** | **91.43±0.76** | **88.14±0.60** | **86.35** |

**Natural Language Generation (NLG).** Across five mainstream LLMs (Llama2, Mistral, RWKV, Qwen3), MiSS consistently achieves the best or near-best average performance (Table 5). Notably, it demonstrates substantial gains in complex reasoning tasks, recording the highest Math score (**34.82**) on Qwen3-4B and the highest average score (**47.79**) on Mistral-7B. These findings highlight that MiSS is not only effective on medium-sized models but also scales robustly to larger architectures and data-rich models.

Table 5: We conduct a systematic comparison of LoRA, DoRA, PiSSA, and MiSS across several mainstream large language models (Llama2, RWKV, Mistral, and Qwen3). All reported results are averaged over three independent runs to ensure robustness. The first-place entry should be highlighted in **bold**, and the second-place entry should be underlined.

| Model | Strategy | Trainable | GSM8K | Math | HumanEval | Mbpp | Avg |
|---|---|---|---|---|---|---|---|
| Llama2-7B (Touvron et al., 2023) | LoRA | 89.9M | 40.75 | 5.22 | 17.74 | 35.15 | 24.72 |
| | DoRA | 91.3M | 42.93 | 6.51 | 21.95 | 36.53 | 26.48 |
| | PiSSA | 89.9M | 43.89 | 6.92 | 22.15 | **37.84** | 27.70 |
| | MiSS | **87.0M** | **48.16** | **8.58** | **23.63** | 36.81 | **29.30** |
| RWKV 6-7B (Peng et al., 2024) | LoRA | 88.1M | 38.13 | 6.06 | - | - | 22.10 |
| | PiSSA | 88.1M | 40.48 | 6.12 | - | - | 23.30 |
| | MiSS | 88.1M | **41.73** | **6.52** | - | - | **24.13** |
| Mistral-7B (Jiang et al., 2023) | LoRA | 94.4M | 62.85 | 15.82 | 35.71 | 46.11 | 40.12 |
| | DoRA | 95.8M | 63.68 | 13.60 | 38.41 | 48.73 | 41.10 |
| | PiSSA | 94.4M | 67.01 | 18.13 | 41.28 | 51.37 | 44.45 |
| | MiSS | **87.0M** | **68.92** | **18.85** | **42.07** | **61.33** | **47.79** |
| Llama2-13B (Touvron et al., 2023) | LoRA | 250M | 56.18 | 12.60 | 31.79 | 37.82 | 34.60 |
| | DoRA | 252M | 61.56 | 13.60 | 33.50 | 39.25 | 36.98 |
| | PiSSA | 250M | 66.64 | 13.82 | 33.57 | 46.03 | 39.52 |
| | MiSS | 255M | **68.64** | **15.74** | **38.15** | **47.91** | **42.11** |
| Qwen3-4B (Yang et al., 2025) | LoRA | 74.3M | 84.38 | 15.20 | 73.27 | 78.32 | 62.79 |
| | DoRA | 75.4M | 85.11 | 21.73 | 74.20 | **78.77** | 64.95 |
| | PiSSA | 74.3M | **85.78** | 26.00 | **75.01** | 78.04 | 66.21 |
| | MiSS | **70.1M** | 85.52 | **34.82** | 74.48 | 78.05 | **68.22** |

**Vision Task** To validate the ability of MiSS to adapt to non-textual tasks, we conducted experiments on the VTAB-1K image and video benchmarks (Table 6). MiSS achieved an average accuracy

of **88.02** on image tasks and **72.96** on video tasks, making it highly competitive with top-performing baseline methods like LoRA and DoRA. Crucially, this competitive performance is delivered with a significantly lower parameter budget ($\approx 0.4$ #TPs) compared to LoRA/DoRA ($\approx 0.8$ #TPs), confirming that the efficiency of MiSS transcends the language domain and is applicable to multimodal foundation models.

Table 6: Performance comparison on VTAB-1K image and video benchmarks.Results are adopted from SliceFine (Kowsher et al., 2025).

| Method | Image | | | | | | | | | Video | | | | |
|---|---|---|---|---|---|---|---|---|---|---|---|---|---|---|
| | Caltech | Flowers | Pets | Camel. | Euro. | Retino. | KITTI | Avg | #TPs | UCF101 | Kinetics | HMDB | Avg | #TPs |
| Full | 89.92 | 97.41 | 85.87 | 81.65 | 88.12 | 73.62 | 77.93 | 84.93 | 85.83 | 92.30 | 55.23 | 65.79 | 74.99 | 86.65 |
| VeRA | 91.53 | 99.19 | 91.04 | 86.45 | 92.97 | 74.25 | 77.92 | 87.62 | 0.240 | 92.28 | 57.21 | 66.77 | 72.09 | 0.242 |
| LoRA | 92.03 | 99.18 | 90.92 | 87.73 | 92.65 | 74.23 | 80.42 | 88.08 | 0.833 | 93.88 | 57.81 | 67.37 | 73.02 | 0.835 |
| DoRA | 91.86 | 99.27 | 91.08 | 85.88 | 91.42 | 75.28 | 80.46 | 87.89 | 0.834 | 92.84 | 57.77 | 67.33 | 72.65 | 0.836 |
| **MiSS** | **92.14** | **99.23** | **91.05** | **86.28** | **92.83** | **73.71** | **80.91** | **88.02** | **0.414** | **93.82** | **57.75** | **67.31** | **72.96** | **0.415** |

## 5.2 EFFECT OF RANK $r$

We evaluate MiSS with varying matrix ranks to study the trade-off between tuning capacity and parameter cost. The Table 7 reports results for ranks $r \in \{16, 32, 64, 128\}$ (corresponding to $\{21.7M, 43.5M, 87.0M, 174.0M\}$ trainable parameters). Performance on GSM8K and the Math benchmark improves monotonically as the rank increases: GSM8K rises from 45.90 at $r = 16$ to 53.49 at $r = 128$, while Math increases from 3.77 to 10.08. In practice, $r = 64$ offers a favorable trade-off (48.16 GSM8K, 8.58 Math) between performance gains and parameter overhead.

Table 7: Comparing different values of rank $(r)$ on LLaMA2-7B with MiSS.

| Model | Rank | Trainable | GSM8K | Math |
|---|---|---|---|---|
| | 16 | 21.7M | 45.90 | 3.77 |
| Llama2-7B | 32 | 43.5M | 46.18 | 7.43 |
| | 64 | 87.0M | 48.16 | 8.58 |
| | 128 | 174.0M | 53.49 | 10.08 |

## 5.3 MISS'S SUPERIOR BALANCE ON THE PARETO FRONTIER: OPTIMALLY TRADING OFF EFFICIENCY AND PERFORMANCE

The emergence of PEFT techniques is motivated by dual objectives: mitigating GPU memory constraints and exploring more efficient model architectures. Nevertheless, numerous contemporary studies disproportionately focus on ultimate performance benchmarks, overlooking critical practical considerations like computational efficiency and training duration—an emphasis that clearly diverges from the original rationale for PEFT. In this section, we undertake a multi-dimensional investigation into the relationships among computational overhead, efficiency, and performance for diverse models. Leveraging the official Hugging Face PEFT (Mangrulkar et al., 2022) benchmarking framework, our evaluations are conducted under fair and reproducible conditions.

The Pareto frontiers in our evaluation provide definitive evidence of MiSS's effectiveness. In every experimental setting, MiSS is uniquely positioned in the top-left corner—the optimal region—signifying that it delivers the best performance with minimal efficiency cost. This consistent advantage underscores MiSS's unique contribution in balancing these competing objectives.



Figure 4: Pareto front of MiSS comparing with other PEFT methods. We select three more methods as the baseline on the balancing of memory and performance.

Table 8: Experimental results across PEFT methods on Llama-3.2-3B.

| PEFT Type | Total Time | Train Time | Test Accuracy | Train Loss | Accelerator Memory (Bytes) | | |
|---|---|---|---|---|---|---|---|
| | | | | | Max | Reserved 99th | Reserved Avg |
| RSLORA | 2069 | 1871 | 0.5299 | 0.5657 | 22,538,092,544 | 17,953,927,987 | 12,128,059,444 |
| C3A | 2125 | 1924 | 0.5102 | 0.5808 | 22,280,142,848 | 17,825,917,829 | 11,804,454,210 |
| MiSS | 1867 | 1664 | 0.5080 | 0.5776 | 20,248,002,560 | 16,303,469,363 | 11,170,837,063 |
| RANDLORA | 2457 | 2213 | 0.5072 | 0.5785 | 22,798,139,392 | 18,436,063,232 | 12,743,670,025 |
| SHIRA | 2085 | 1867 | 0.5072 | 0.5789 | 21,743,271,936 | 17,637,383,864 | 12,240,924,809 |
| OFT | 2494 | 2214 | 0.5057 | 0.5947 | 22,294,822,912 | 17,939,310,837 | 12,057,354,384 |
| LORA | 1993 | 1796 | 0.4822 | 0.6069 | 22,273,851,392 | 17,710,763,212 | 11,868,689,976 |
| DORA | 2287 | 2023 | 0.4807 | 0.6068 | 24,553,455,616 | 19,189,150,515 | 12,490,471,636 |
| LORAFA | 2026 | 1821 | 0.4299 | 0.6510 | 20,187,185,152 | 16,257,394,933 | 11,106,307,276 |
| LOHA | 2591 | 2341 | 0.4185 | 0.6570 | 23,886,561,280 | 19,247,870,771 | 13,446,820,344 |
| IA3 | 1922 | 1746 | 0.4124 | 0.6569 | 23,135,780,864 | 18,398,356,439 | 12,023,331,867 |
| ADALORA | 2209 | 1986 | 0.3904 | 0.6863 | 22,793,945,088 | 18,203,426,160 | 12,361,399,900 |
| LOKR | 2352 | 2152 | 0.3753 | 0.6877 | 23,565,697,024 | 18,987,698,094 | 13,173,683,073 |
| P_TUNING | 1918 | 1707 | 0.3707 | 0.6740 | 20,937,965,568 | 17,215,688,540 | 11,867,101,593 |
| VBLORA | 2210 | 1962 | 0.3700 | 0.7143 | 22,181,576,704 | 17,635,223,797 | 11,735,344,663 |
| VERA | 2025 | 1820 | 0.3685 | 0.6927 | 21,596,471,296 | 17,291,123,097 | 11,489,715,316 |
| BOFT | 11,114 | 8292 | 0.3647 | 0.7268 | 24,427,626,496 | 20,103,445,872 | 14,814,855,089 |
| IA3 | 2005 | 1783 | 0.3450 | 0.7657 | 23,137,878,016 | 18,398,566,154 | 12,023,227,429 |
| TRAINABLE_TOKENS | 1814 | 1572 | 0.2881 | 0.7862 | 20,956,839,936 | 16,957,675,929 | 12,730,137,942 |
| PROMPT_TUNING | 2715 | 2394 | 0.2525 | 0.7790 | 24,408,752,128 | 20,650,676,715 | 15,297,364,466 |
| ADAPTION_PROMPT | 2261 | 1989 | 0.2206 | 0.8317 | 22,410,166,272 | 17,907,664,814 | 11,893,757,234 |
| PREFIX_TUNING | 1959 | 1662 | 0.1471 | 0.7887 | 20,912,799,744 | 16,945,051,074 | 11,766,684,083 |
| FOURIERFT | 2824 | 2422 | 0.1198 | 0.9979 | 23,681,040,384 | 19,054,869,872 | 13,111,221,498 |
| PROMPT_TUNING | 2700 | 2380 | 0.0500 | 1.0655 | 24,379,392,000 | 20,669,781,770 | 15,297,773,830 |
| FOURIERFT | 2824 | 2424 | 0.0008 | 1.2480 | 23,653,777,408 | 19,017,267,937 | 13,104,129,350 |
| LN_TUNING | 1870 | 1657 | 0.0000 | 1.2370 | 21,177,040,896 | 16,903,066,091 | 11,385,589,622 |

# 6  CONCLUSION

This work tackles the critical inefficiency of simultaneous matrix updates in Low-Rank Adaptation (LoRA), which leads to slow convergence and suboptimal resource use. We propose MiSS as a compelling solution—a new PEFT framework that updates decomposed weight shards using a single, shared matrix. This approach drastically reduces optimization complexity and resource demands. Comprehensive experiments validate that MiSS consistently outperforms existing methods in accuracy, memory footprint, and computational speed, offering a fundamentally more efficient pathway for adapting large models.

# 7  LIMITATIONS AND FUTURE WORK

As a pioneering approach, MiSS still leaves several aspects open for deeper exploration. We hope that future research will conduct broader and more in-depth studies to further refine PEFT techniques and identify the most effective strategies for large language models.

# 8  ACKNOWLEDGMENTS

This is the first paper of my life, and I am sincerely grateful to those who have supported me along the way.

I would like to express my heartfelt thanks to Huanqi Cao, Cormen, Yu Zhang, Jiaming Kong, and Houhao Wen. Although their names do not appear on this paper, this work would not have been possible without their guidance, discussions, and encouragement. I have benefited greatly from their insights and generosity. I am also deeply thankful to everyone who has guided and inspired me throughout my journey. Even if not formally acknowledged as authors, their influence is reflected in this work.

I hope to continue pursuing research that is both meaningful and interesting in the years to come.

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

## A APPENDIX

### A.1 ADDITIONAL EXPERIMENTS

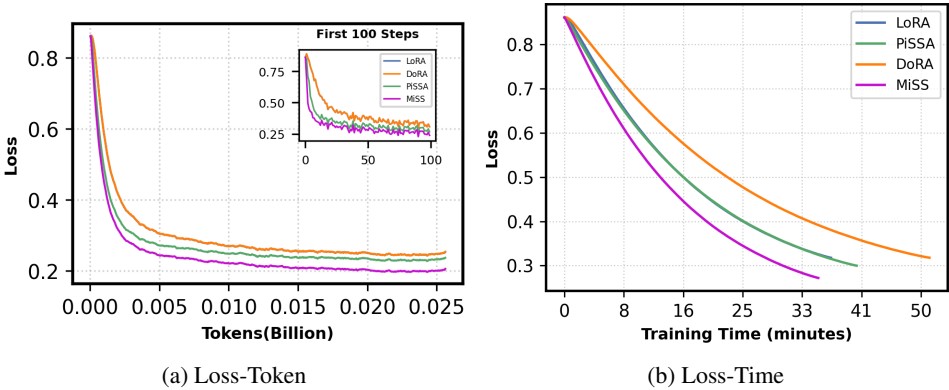

(a) Loss-Token  (b) Loss-Time

Figure 5: Loss curves of LLaMA2-7B fine-tuned on MetaMathQA using LoRA and MiSS(a) Loss vs. tokens. (b) Loss vs. training time.

### A.2 RWKV7

Table 9: We fine-tuned LLMs using MiSS and various LoRA variants, and evaluated performance on GSM8k, Math, HumanEval, and MT-Bench.

| Model | Strategy | Trainable | GSM8K | Math | HumanEval | MT-Bench |
|---|---|---|---|---|---|---|
| | Base | 0M | 44.35 | - | - | - |
| RWKV7-3B | LoRA | 47.2M | 55.64 | - | - | - |
| | PiSSA | 47.2M | 57.16 | - | - | |
| | MiSS | 47.2M | **58.22** | - | - | - |

## B SETTINGS OF EXPERIMENTS

**NLU** We fine-tune the RoBERTa-base model on several datasets from the GLUE benchmark, including MNLI, SST-2, CoLA, QNLI, and MRPC. Performance is evaluated on the development set using accuracy as the primary metric. The experimental hyperparameter settings were aligned with those in the LoRA repository, but training was conducted using a single 4090 GPU. Each experiment is conducted with 3 different random seeds, and the average performance is reported. As shown in Table 4, MiSS demonstrates outstanding performance, particularly on the CoLA dataset, where it exhibits significantly faster convergence and superior data-fitting capabilities, far surpassing LoRA and PiSSA.

Table 10: Hyperparameter settings for fine-tuning llama2-7B,Mistral-7B,RWKV6-7B,Qwen3-4B on NLG tasks

| Hyperparameters | LoRA | DoRA | PiSSA | MiSS |
|---|---|---|---|---|
| Rank r | 36 | 36 | 36 | 64 |
| $\alpha$ | 72 | 72 | 36 | - |
| Dropout | | | 0.0 | |
| Optimizer | | | AdamW | |
| LR | | | 2e-5 | |
| LR Scheduler | | | Cosine decay | |
| Batch size | | | 64 | |
| Warmup ratio | | | 0.0 | |
| Epochs | | | 1 | |
| Where | | | Q,K,V,O,Up,Down,Gate | |

Table 11: Hyperparameter settings for fine-tuning llama2-13B on NLG tasks

| Hyperparameters | LoRA | DoRA | PiSSA | MiSS |
|---|---|---|---|---|
| Rank r | 64 | 64 | 64 | 128 |
| $\alpha$ | 128 | 128 | 64 | - |
| Dropout | | | 0.0 | |
| Optimizer | | | AdamW | |
| LR | | | 2e-5 | |
| LR Scheduler | | | Cosine decay | |
| Batch size | | | 128 | |
| Warmup ratio | | | 0.0 | |
| Epochs | | | 1 | |
| Where | | | Q,K,V,O,Up,Down,Gate | |

**NLG**   To verify the generalizability of MiSS, we conducted more comprehensive experiments on LLM. we conducted 3 more task finetuning experiments on LLM: *math* and *code*. (1) *Math*: We trained our model on a 395k subset of MetaMathQA (Yu et al., 2023), a dataset bootstrapped from other math instruction tuning datasets like GSM8K (Cobbe et al., 2021) and MATH (Yu et al., 2023), with higher complexity and diversity. (2) *Code*: We train our model on a 100k subset of CodeFeedback (Zheng et al., 2024), a high-quality code instruction dataset, removing explanations after code blocks. The model is tested on HumanEval (Chen et al., 2021) and Mbpp (Austin et al., 2021). The hyperparameter settings for this experiment were kept equal, while the train steps were adjusted according to the specific fine-tuning datasets used. It is worth noting that the attention-based architectures employed by models such as LLaMA, Qwen, and Mistral do not use fully symmetric weight structures, which makes it impossible to achieve exact alignment of trainable parameters when comparing MiSS with LoRA. To address this, we set the rank $r$ of LoRA to 36 and the rank $r$ of MiSS to 64, ensuring that MiSS uses fewer parameters than LoRA to demonstrate its superiority. Each experiment is conducted with 2 different random seeds, and the average performance is reported.

**Vision Task**   on VTAB-1K image classification using ViT-Base-Patch16-224

## C   SETTINGS OF EXPERIMENTS IN NO FREE LUNCH

Table 12: Experimental Setup: Datasets and Hyperparameters

| General Configuration | |
|---|---|
| Parameter | Value |
| Random Seed (SEED) | 43 |
| Device (DEVICE) | CUDA (if available, else CPU) |
| **Base Model Architecture (MLP)** | |
| Input Dimension | 64 |
| Hidden Dimension | 64 |
| Output Dimension | 64 |
| **Synthetic Dataset Generation** | |
| Base Function | $\sin(2\pi x)$ |
| Modified Function | $\sin(2\pi x) + 0.3\cos(3\pi x)$ |
| Input $x$ Range | $[-1, 1]$ |
| Training Samples ($N\_TRAIN$) | 50 |
| Validation Samples ($N\_VALID$) | 100 |
| Training Noise Std. Dev. (NOISE_STD) | 0.05 |
| Validation Noise Std. Dev. | 0.0 |
| **Training Parameters** | |
| Base Model LR (BASE_LR) | 0.001 |
| Adaptation LR (ADAPT_LR) | 0.001 |
| Base Model Epochs (BASE_EPOCHS) | 250 |
| Adaptation Epochs (ADAPT_EPOCHS) | 100 |
| Evaluation Interval (EVAL_INTERVAL) | 10 |
| **Adapter-Specific Ranks** | |
| LoRA Rank | 2 |
| VeRA Rank | 64 |
| MiSSRank | 4 |
| PiSSA Rank | 2 |
| DoRA Rank | 1 |
| ProLoRA Rank | 2 |
| AdaLoRA Rank | 2 |
| MoS Rank | 2 |

**Note:** Other adapter-specific hyperparameters (e.g., LoRA scale, VeRA d_init_val, DoRA lora_alpha, ProLoRA unshared_rank_u, MoS shard_dim_ratio) primarily use their default values as defined in the respective adapter class implementations or are derived based on the rank within benchmark functions. Refer to the provided Python code for their specific configurations during experiments.

