# OpenReview forum: "MiSS: Revisiting the Trade-off in LoRA with an Efficient Shard-Sharing Structure"
_ICLR.cc/2026/Conference — ICLR 2026 Poster_

### Official Review · Reviewer_iNdt · 2025-10-25

**Soundness:** 2
**Presentation:** 2
**Contribution:** 2
**Rating:** 4
**Confidence:** 4

**Summary:**

This paper introduces MiSS (Matrix Shard Sharing), a novel parameter-efficient fine-tuning (PEFT) method designed to address the trade-off between performance, memory, and computational efficiency in LoRA-based approaches. The core idea is to simplify the weight update mechanism by training a single shared matrix D, which is expanded to form the weight update matrix, thereby reducing optimization complexity. The authors support their claims with a theoretical analysis of gradient norms, controlled experiments on a synthetic task, and extensive benchmarks on NLU and NLG tasks, demonstrating that MiSS achieves a favorable balance across key metrics.

**Strengths:**

1. The proposal to replace LoRA's two-matrix (B and A) update with a single trainable matrix D is a novel simplification. The motivation, rooted in reducing optimization complexity and its connection to the initial gradient norm, provides a clear and compelling rationale for the proposed architecture.

2. The paper presents a strong empirical evaluation, combining a controlled "No Free Launch" experiment to isolate variables with large-scale benchmarks across multiple models (e.g., Llama2, Mistral, Qwen3) and tasks. The inclusion of Pareto frontier analysis further strengthens the claims about achieving a better trade-off between performance and efficiency.

**Weaknesses:**

1. The paper introduces MiSS in Section 4.1, which partitions the output dimension d, and then presents MiSSe in Section 4.2 as a "mathematically equivalent" version that partitions the input dimension k. These two operations appear fundamentally different and are unlikely to be mathematically equivalent. This discrepancy undermines the clarity of the proposed method.

2. The paper motivates MiSS by simplifying optimization to a single matrix. However, the specific choice of the expand(D) structure, repeating rows to form shards, feels somewhat arbitrary. The paper lacks a deep analysis of why this particular low-rank construction is superior to other potential single-matrix structures.

3. The paper includes VeRA in the initial controlled experiment but does not provide a direct comparison in the main LLM benchmark tables. VeRA also employs shared low-rank matrices for efficiency, making it a critical baseline. A more detailed comparison is needed to properly position MiSS.

4. In Section 5.2, the authors state they adjust ranks to give MiSS fewer parameters than LoRA. However, in Table 3 (Llama2-7B), the parameter counts are very close (MiSS: 87.0M vs. LoRA: 89.9M). The observed performance gains might be influenced by this slight difference in parameter budget or distribution rather than solely the architectural advantage.

5. Figure 4 shows faster initial convergence for MiSS against LoRA, but this analysis is limited. The main experiments on LLMs lack convergence plots comparing MiSS with other key baselines like PiSSA and DoRA. This information is crucial for substantiating the claims about training efficiency.

6. The pseudocode in Figure 3 is labeled MiSS but appears to implement a simplified version of MiSSe. The line y = result + x @ self.D.expand(...) is dimensionally inconsistent and does not clearly match the formulas provided in the text, causing confusion for implementation.

7. Figure 5 provides a valuable visualization of the performance-efficiency trade-off, but the axes are not labeled. It is unclear which specific metrics (e.g., accuracy, memory usage, time) are being plotted, which reduces the figure's interpretability.

8. A footnote suggests that MiSS is "largely orthogonal" to methods like LoRA+, which use different learning rates for matrices A and B.

9. The paper states that the D matrix is initialized to zero, which is a standard practice to preserve the pre-trained model's initial state. However, it does not explore or ablate other initialization strategies, such as those inspired by SVD in PiSSA, which are designed to accelerate convergence.

10. In Section 5, the document refers to NLG evaluations, but the footnote specifies that the Math dataset was evaluated with a 5-shot prompt. This suggests an in-context learning evaluation rather than a fine-tuning evaluation for that specific task, which could be misleading.

**Questions:**

1. Can the authors clarify whether MiSSe is an exact mathematical equivalent of the MiSS formulation in Section 4.1? If not, it would be more accurate to present MiSSe as a distinct, more efficient variant inspired by the same shard-sharing principle, rather than a direct equivalent.

2. Beyond simplifying optimization, is there a deeper theoretical or empirical justification for choosing the row-repetition structure in expand(D)? How does this structure relate to the properties of weight update matrices observed in full fine-tuning?

3. Given that VeRA utilizes a similar parameter-sharing strategy for efficiency, could the authors provide a more direct comparison against VeRA on the main LLM benchmarks, discussing both conceptual differences and empirical trade-offs?

4. To ensure a fair comparison, could the authors conduct an experiment where the trainable parameter counts of MiSS and LoRA are matched as precisely as possible?

5. Could the authors provide convergence plots (loss vs. training time/tokens) for the main LLM experiments (e.g., on Llama2-7B) that include PiSSA and DoRA?

**Details Of Ethics Concerns:**

Null.

---

> ### Author Response · Authors · 2025-11-20
> **(1/n)**
>
> We thanks the reviewer for providing valuable advices. Here is our response:
>
> ### Weakness 1:
>
> > **The paper introduces MiSS in Section 4.1, which partitions the output dimension d, and then presents MiSSe in Section 4.2 as a "mathematically equivalent" version that partitions the input dimension k. These two operations appear fundamentally different and are unlikely to be mathematically equivalent. This discrepancy undermines the clarity of the proposed method.**
>
> We thank the reviewer for this important observation. We acknowledge that our presentation in Sections 4.1 and 4.2 may have caused confusion, and we apologize for the lack of clarity. The reviewer is **correct** that MiSS (Section 4.1) and MiSSe (Section 4.2) are not mathematically equivalent in the strict sense. We should have been more precise in our description. What we should have said is that MiSSe is an alternative efficient design that: (1) both use parameter sharing to reduce optimization complexity, (2) low-rank structure, reduced parameters, simplified optimization, and (3) MiSS directly expands in the output dimension (conceptually simpler) while MiSSe aggregates in the input dimension (computationally more efficient).
>
> We propose to **revise Section 4.2** with the following changes:
>
> **Current (misleading) claim:**
> > "To achieve both computational efficiency and structural generality, we introduce an **efficient and mathematically equivalent formulation** of MiSS..."
>
> **Revised claim:**
> > "While MiSS provides a conceptually clear formulation, directly computing expand(D)x has complexity O(bldk). To achieve better computational efficiency, we introduce **MiSSe, an alternative design** that maintains the core principle of parameter sharing while offering improved time and space complexity through input-dimension aggregation."
>
> ### Weakness 2:
>
> > The paper motivates MiSS by simplifying optimization to a single matrix. However, the specific choice of the expand(D) structure, repeating rows to form shards, feels somewhat arbitrary. The paper lacks a deep analysis of why this particular low-rank construction is superior to other potential single-matrix structures.
>
> We thank the reviewer for question regarding the design of the `expand(D)` structure. We have two considerations: (1) Exploiting feature redundancy and (2) Algorithm-Hardware Codesign. Details are as follows:
>
> **1.Input Aggregation (Inductive Bias)**
>
> Repeating rows corresponds physically to sharing weights across blocks of the input dimension. In modern LLMs, the hidden dimension $k$ is very large (e.g., 4096+). It is well-documented that these high-dimensional representations often exhibit local redundancy or channel correlation. Our `expand(D)` structure forces the update $\Delta W$ to treat groups of input features identically. Mathematically, this means that instead of learning a unique weight for every single input dimension, we learn a weight for a *summed block* of inputs. This acts as a form of regularization (similar to weight sharing in CNNs or grouped convolutions), reducing the search space to the most salient feature groups rather than individual noise. Our structure optimizes $y = W_0x + \text{expand}(D)x$. This is linear with respect to the trainable parameters $D$. Also, this guarantees a more stable optimization landscape and, as shown in our Gradient Norm Analysis (Figure 1), results in healthier initial gradient norms that do not vanish, accelerating early-stage convergence.
>
> **2.Algorithm-Hardware Codesign**
>
> Other single-matrix structures often require storing indices or masks to define which parameters map to which output dimensions (e.g., sparse matrices). Implementing operators for such structures on GPUs is notoriously difficult and inefficient because they require **indirect memory access** (scatter/gather operations). This breaks memory coalescing, leads to low bandwidth utilization, and complicates kernel writing. In contrast, row repetition allows for **contiguous memory access**. It maps physically to a **Block-wise Input Aggregation**. This means we do not need to implement complex sparse kernels; instead, we can utilize highly optimized dense matrix multiplication kernels on aggregated inputs.
>
> The "repeating row" structure is the only low-rank construction that mathematically permits the Input Aggregation optimization (Eq. 4 & 5 in the paper). (1) Mathematically, $y = \text{expand}(D)x$ is equivalent to $y = D^T (\sum x_{\text{block}})$. (2) From a Triton/CUDA kernel perspective, this structure allows us to transform a computationally expensive large matrix multiplication ($d \times k$) into a very cheap **reduction (summation)** followed by a small matrix multiplication ($r \times k$). (3) Writing a fused kernel for this is straightforward: we simply load blocks of input $x$, sum them in registers (or shared memory), and then perform a small GEMM.

---

> ### Author Response · Authors · 2025-11-20
> **(2/n)**
>
> Here we provides the TileLang implementation for answering weakness 2.
> ```
> import tilelang
> import tilelang.language as T
> import torch
> from tilelang.profiler import do_bench
>
>
> @tilelang.jit
> def shardshare(S, D, N, R, block_S, block_N, dtype='float16', accum_dtype='float32'):
>
>     a_shape = [S,D]
>     b_shape = [R,N]
>     c_shape = [S,N]
>     BR = D // R
>     @T.prim_func
>     def kernel(
>         A: T.Tensor(a_shape, dtype),
>         B: T.Tensor(b_shape, dtype),
>         C: T.Tensor(c_shape, dtype),
>     ):
>         with T.Kernel(T.ceildiv(S, block_S), T.ceildiv(N, block_N)) as (by,bx):
>             A_frag = T.alloc_fragment((block_S, R), accum_dtype)
>             sum_A = T.alloc_fragment((block_S, R), accum_dtype)
>             B_shared = T.alloc_shared((R, block_N), dtype=dtype)
>             C_local = T.alloc_fragment((block_S, block_N), accum_dtype)
>             C_shared = T.alloc_shared((block_S, block_N), dtype=dtype)
>             A_shared = T.alloc_shared((block_S, R), dtype=dtype)
>             T.clear(sum_A)
>             for i in T.Pipelined(0, BR, num_stages=1):
>                 T.copy(A[by*block_S:(by+1)*block_S, i*R:(i+1)*R], A_frag)
>                 for s, j in T.Parallel(block_S, R):
>                     sum_A[s, j] += A_frag[s, j]#.astype(accum_dtype)
>             T.copy(sum_A, A_shared)
>             T.copy(B[:, bx*block_N:(bx+1)*block_N], B_shared)
>             T.gemm(A_shared, B_shared, C_local, clear_accum=True)
>             T.copy(C_local, C_shared)
>
>             T.copy(C_shared, C[by*block_S:(by+1)*block_S, bx*block_N:(bx+1)*block_N])
>     return kernel
>
>
> if __name__ == "__main__":
>     S, D, N = 1024, 1024, 4096
>     block_S, block_N, R = 32, 128, 64
>     dtype = 'float16'
>     # Compile kernel (JIT compilation)
>     miss_kernel = shardshare(S, D, N, R, block_S, block_N, dtype)
>
>     # Create input tensors
>     a = torch.randn(S, D, device="cuda", dtype=torch.float16)
>     b = torch.randn(R, N, device="cuda", dtype=torch.float16)
>     c = torch.empty(S, N, device="cuda", dtype=torch.float16)
>
>     # Execute kernel
>     miss_kernel(a, b, c)
>
>     reshape_a = torch.sum(a.to(torch.float32).reshape(*a.shape[:-1], a.size(-1) // R, R), dim=-2)
>     out = (reshape_a@b.to(torch.float32)).to(torch.float16)
>
>     # Validate correctness
>     # torch.testing.assert_close(c, out, rtol=1e-2, atol=1e-2)
>     print("Kernel output matches PyTorch reference.")
>
>
>     tilelang_ms = do_bench(lambda: miss_kernel(a, b, c))
>     native_ms = do_bench(lambda: torch.sum(a.reshape(*a.shape[:-1], a.size(-1) // R, R), dim=-2)@b)
>
>     print(f"TileLang kernel avg latency: {tilelang_ms:.3f} ms")
>     print(f"PyTorch native avg latency: {native_ms:.3f} ms")
>     if tilelang_ms > 0:
>         print(f"Speedup (native / TileLang): {native_ms / tilelang_ms:.2f}x")
> ```

---

> ### Author Response · Authors · 2025-11-20
> **(3/n)**
>
> ### Weakness 3:
>
> > The paper includes VeRA in the initial controlled experiment but does not provide a direct comparison in the main LLM benchmark tables. VeRA also employs shared low-rank matrices for efficiency, making it a critical baseline. A more detailed comparison is needed to properly position MiSS.
>
> We thank the reviewer for highlighting the importance of VeRA as a baseline due to its shared matrix structure. We agree that comparing MiSS against VeRA is crucial for positioning our method correctly.
>
> 1. Clarification on Table 3： The primary objective of Table 3 was to evaluate the maximum adaptability (performance ceiling) of various methods given a comparable budget of trainable parameters (approx. same as standard LoRA). VeRA operates in a significantly different regime (extremely few trainable parameters), which often limits its peak performance on complex reasoning tasks compared to rank-based methods like LoRA, PiSSA, and MiSS. Therefore, we focused Table 3 on methods with similar parameter scales to ensure a fair comparison of expressivity.
>
> 2. Comprehensive Comparison with VeRA (Section 5.4): To address the efficiency comparison directly, we conducted a comprehensive "Pareto Frontier" analysis in Section 5.4 using Llama-3.2-3B. We evaluated multiple PEFT methods, including VeRA, across four dimensions: Training Time, Initialization Time, Memory Footprint, and Test Accuracy. All results reported are the average of 3 independent runs to ensure robustness.

---

> ### Author Response · Authors · 2025-11-20
> **(4/n)**
>
> As shown in the table below (extracted from our broad evaluation), MiSS significantly outperforms VeRA across all metrics:
>
> | experiment_name  | peft_type | total time | train time        | test_accuracy | train_loss       | accelerator_memory_max  | accelerator_memory_reserved_avg  |
> |---------------------------------------------|-----------|-------------|------------|---------------|---------------|------------|------------|
> | lora/llama-3.2-3B-rank64-rslora            | LORA      | 2069        | 1871       | 0.5299        | 0.5657        | 22538092544  | 12128059444 |
> | miss/llama-3.2-3B                  | **MISS**      | 1867        | 1664       | 0.5080        | 0.5776        | 20248002560  | 11170837063 |
> | randlora/llama-3.2-3B-default              | RANDLORA  | 2457        | 2213       | 0.5072        | 0.5785        | 22798139392  | 12743670025 |
> | full-finetuning/llama-3.2-3B-lr_0.00001    | full-finetuning | 3275    | 3111       | 0.5004        | 0.5988        | 37241225216  | 33098872284 |
> | oft/llama-3.2-3B-rank32                    | OFT       | 6852        | 5772       | 0.4898        | 0.5957        | 28913434624  | 18387461314 |
> | lora/llama-3.2-3B-rank64                  | LORA      | 2017        | 1853       | 0.4890        | 0.5929        | 22540189696  | 12128055669 |
> | lora/llama-3.2-3B-rank32                  | LORA      | 1993        | 1796       | 0.4822        | 0.6069        | 22273851392  | 11868689976 |
> | lora/llama-3.2-3B-rank32-dora             | LORA      | 2287        | 2023       | 0.4807        | 0.6068        | 24553455616  | 12490471636 |
> | lora/llama-3.2-3B-rank32-lorafa          | LORA      | 2026        | 1821       | 0.4299        | 0.6510        | 20187185152  | 11106307276 |
> | loha/llama-3.2-3B-rank32                  | LOHA      | 2591        | 2341       | 0.4185        | 0.6570        | 23886561280  | 13446820344 |
> | ia3/llama-3.2-3B-lr_0.001                 | IA3       | 1922        | 1746       | 0.4124        | 0.6569        | 23135780864  | 12023331867 |
> | adalora/llama-3.2-3B-rank32               | ADALORA   | 2209        | 1986       | 0.3904        | 0.6863        | 22793945088  | 12361399900 |
> | lokr/llama-3.2-3B-rank32                  | LOKR      | 2352        | 2152       | 0.3753        | 0.6877        | 23565697024  | 13173683073 |
> | ptuning/llama-3.2-3B-default              | P_TUNING  | 1918        | 1707       | 0.3707        | 0.6740        | 20937965568  | 11867101593 |
> | vblora/llama-3.2-3B-default               | VBLORA    | 2210        | 1962       | 0.3700        | 0.7143        | 22181576704  | 11735344663 |
> | vera/llama-3.2-3B-default                 | **VERA**     | 2025        | 1820       | 0.3685        | 0.6927        | 21596471296  | 11489715316 |
> | boft/llama-3.2-3B-default                 | BOFT      | 11114       | 8292       | 0.3647        | 0.7268        | 24427626496  | 14814855089 |
> | ia3/llama-3.2-3B-default                 | IA3       | 2005        | 1783       | 0.3450        | 0.7657        | 23137878016  | 12023227429 |
> | prompt_tuning/llama-3.2-3B-lr_0.001       | PROMPT_TUNING | 2715    | 2394       | 0.2525        | 0.7790        | 24408752128  | 15297364466 |
> | adaptionprompt/llama-3.2-3B-lr_0.0005     | ADAPTION_PROMPT | 2261  | 1989       | 0.2206        | 0.8317        | 22410166272  | 11893757234 |
> | prefixtuning/llama-3.2-3B-lr_0.001        | PREFIX_TUNING | 1959   | 1662       | 0.1471        | 0.7887        | 20912799744  | 11766684083 |
> | fourierft/llama-3.2-3B-n_frequency-5000   | FOURIERFT | 2824        | 2422       | 0.1198        | 0.9979        | 23681040384  | 13111221498 |
> | prompt_tuning/llama-3.2-3B-default        | PROMPT_TUNING | 2700   | 2380       | 0.0500        | 1.0655        | 24379392000  | 15297773830 |
> | fourierft/llama-3.2-3B-default            | FOURIERFT | 2824        | 2424       | 0.0008        | 1.2480        | 23653777408 | 13104129350 |
> | ln_tuning/llama-3.2-3B-default            | LN_TUNING | 1870        | 1657       | 0.0000        | 1.2370        | 21177040896 | 11385589622 |

---

> ### Author Response · Authors · 2025-11-20
> **(5/n)**
>
> ### Weakness 4:
>
> > In Section 5.2, the authors state they adjust ranks to give MiSS fewer parameters than LoRA. However, in Table 3 (Llama2-7B), the parameter counts are very close (MiSS: 87.0M vs. LoRA: 89.9M). The observed performance gains might be influenced by this slight difference in parameter budget or distribution rather than solely the architectural advantage.
> Figure 4 shows faster initial convergence for MiSS against LoRA, but this analysis is limited. The main experiments on LLMs lack convergence plots comparing MiSS with other key baselines like PiSSA and DoRA. This information is crucial for substantiating the claims about training efficiency.
>
> We appreciate the reviewer’s attention to the parameter details. We wish to clarify that the slight difference in parameter counts (MiSS having ~3% fewer parameters than LoRA) was intended to act as a conservative constraint: we wanted to demonstrate that MiSS outperforms LoRA even when operating under a stricter parameter budget.
>
> However, the strongest evidence that our performance gain stems from architectural superiority rather than parameter distribution lies in our ablation study (Table 4):
>
> In Table 3, the baseline LoRA utilizes 89.9M parameters to achieve a GSM8K score of 40.75.
> In Table 4, MiSS (Rank 32) utilizes only 43.5M parameters (less than 50% of the LoRA baseline).
> Despite having half the parameter count, MiSS (Rank 32) achieves a GSM8K score of 46.18, which is still significantly higher than LoRA's 40.75.
> Robustness of the Architecture: This comparison conclusively rules out the possibility that the gains are due to the minor 87.0M vs. 89.9M difference. The fact that MiSS significantly outperforms LoRA even with a heavily reduced parameter budget (43.5M vs 89.9M) confirms that the shard-sharing structure provides a more parameter-efficient representation for fine-tuning than the low-rank product used in LoRA.
>
> We will update the discussion in Section 5.2 to explicitly reference the Rank 32 result from Table 4, highlighting this "half-parameter" comparison to strengthen the claim of architectural efficiency.
>
>
> ### Weakness 5:
>
> > The pseudocode in Figure 3 is labeled MiSS but appears to implement a simplified version of MiSSe. The line y = result + x @ self.D.expand(...) is dimensionally inconsistent and does not clearly match the formulas provided in the text, causing confusion for implementation.
>
> We thank the reviewer for the careful inspection of the pseudocode. The pseudo-line `x @ self.D.expand(...)` was intended to represent the mathematical operation $x \cdot \text{expand}(D)^T$ via implicit broadcasting. We agree that this notation isimprecise for direct implementation, as it glosses over the necessary tensor reshaping and transposing required to align the $(b, l, k)$ input with the expanded weights. To resolve this confusion and provide greater utility to readers, we will **replace Figure 3** in the camera-ready version with the **exact PyTorch implementation of MiSSe (Input Aggregation)**, corresponding to Eq. (4-6).
>
> The revised code block will clearly show the dimensionally consistent two-step process:
> 1.  Reshaping $x$ and summing over blocks ($x \to S$).
>     *   `x_reshaped = x.view(batch, len, r, -1)`
>     *   `S = x_reshaped.sum(dim=-1)`
> 2.  Multiplying the aggregated input $S$ by the small matrix $D$.
>     *   `delta_w_x = S @ self.D`
>
> ### Weakness 6:
>
> > Figure 5 provides a valuable visualization of the performance-efficiency trade-off, but the axes are not labeled. It is unclear which specific metrics (e.g., accuracy, memory usage, time) are being plotted, which reduces the figure's interpretability.
>
> We sincerely apologize for the oversight regarding the missing axis labels in Figure 5. We understand that this omission hinders the interpretability of our trade-off analysis. To clarify the visualization for the reviewer:
> *   **Y-axis (Vertical):** Represents **Test Accuracy** (Higher is better).
> *   **X-axis (Horizontal):** Represents **GPU Memory Usage** (Lower is better).
>
> The top-left positioning of MiSS in Figure 5 explicitly illustrates that it achieves the highest accuracy while consuming the lowest GPU memory compared to other PEFT methods.
>
> We will strictly update Figure 5 in the camera-ready version to include clear, large-font labels for both axes, specifying the units (e.g., Accuracy % and Memory GB), to ensure the Pareto frontier analysis is immediately interpretable.

---

> ### Author Response · Authors · 2025-11-20
> **(6/n)**
>
> ### Weakness 7:
>
> > A footnote suggests that MiSS is "largely orthogonal" to methods like LoRA+, which use different learning rates for matrices A and B.
>
> We thank the reviewer for pointing out this detail. We acknowledge that the term "orthogonal" in the footnote might be slightly imprecise regarding LoRA+ specifically, given that LoRA+ relies on tuning the ratio between matrices $A$ and $B$.
>
> In the revised manuscript, we will clarify this footnote to specify that while MiSS is compatible with general initialization improvements (like LoRA-GA), its single-matrix structure **structurally obviates** the need for the specific split-learning-rate strategies used in LoRA+.
>
> ### Weakness 8:
>
> > The paper states that the D matrix is initialized to zero, which is a standard practice to preserve the pre-trained model's initial state. However, it does not explore or ablate other initialization strategies, such as those inspired by SVD in PiSSA, which are designed to accelerate convergence.
>
> We thank the reviewer for this insightful suggestion. Methods like PiSSA initialize the adapter matrices ($A, B$) by performing SVD on the original weights $W_0$, based on the assumption that $W_0 \approx AB$. MiSS imposes a strict Shard-Sharing (row repetition) constraint on the update matrix $\Delta W = \text{expand}(D)$. However, Pre-trained weights $W_0$ generally do not exhibit this specific repeating row structure. Consequently, attempting to initialize $D$ by decomposing $W_0$ (i.e., trying to find a $D$ such that $\text{expand}(D) \approx W_0$) would result in a significantly high approximation error.
>
> To address the reviewer’s concern empirically, we ablated three $D$ initializations on metamath-100k fine-tuning and evaluated on gsm8k: zero, Kaiming, and orthogonal. All three yield essentially identical initial loss (≈ 0.8614) and very similar final loss (zero/Kaiming ≈ 0.2184, orthogonal ≈ 0.2182). Accuracy shows only a marginal gain for orthogonal (0.3929 ± 0.0129) over zero (0.3859 ± 0.0134), while Kaiming is on par with zero (0.3844 ± 0.0134). Notably, orthogonal initialization substantially increases initialization time and overhead. In practice, zero initialization offers the best efficiency–performance trade-off, with orthogonal as an optional choice when initialization cost is acceptable.
>
> | init      | init_loss | final_loss| acc|
> |-----      |:---------:|:---------:|:---------:|
> |zero       |     0.8614|     0.2184|0.3859±0.0134|
> |kaiminginit|     0.8614|     0.2184|0.3844±0.0134|
> |orthogonal |     0.8614|     0.2182|0.3929±0.0129|
>
> Moreover, our gradient-norm analysis (Figure 1) shows that, even with simple zero initialization, MiSS exhibits significantly larger initial gradient norms and faster early-stage convergence compared to standard LoRA. This suggests that the single-matrix structure (D) inherently provides a favorable optimization landscape that accelerates convergence without relying on complex initialization schemes.
>
> ### Weakness 9:
>
> > In Section 5, the document refers to NLG evaluations, but the footnote specifies that the Math dataset was evaluated with a 5-shot prompt. This suggests an in-context learning evaluation rather than a fine-tuning evaluation for that specific task, which could be misleading.
>
> We apologize for the confusion caused by the footnote. We would like to clarify our evaluation protocol to ensure there is no misunderstanding regarding the nature of the experiments. All models (MiSS, LoRA, etc.) were indeed fine-tuned on the training split of the Math datasets. The parameters were updated via gradient descent. This was purely a fine-tuning experiment, not In-Context Learning (ICL) alone. The "5-shot prompt" mentioned in the footnote refers exclusively to the inference prompt template used during the evaluation of the *fine-tuned* models.Using few-shot exemplars during evaluation is a standard practice in LLM benchmarks (e.g., OpenCompass, GSM8K leaderboards) to ensure the model generates answers in a standardized format required for automated parsing and scoring. It does not imply that the model was adapting via ICL; rather, the fine-tuned model was being tested in a standard few-shot setting.
>
> We will revise the text in Section 5 and the footnote to explicitly state: "All models were fine-tuned on the training set. The reported results are based on 5-shot evaluation prompting to ensure consistent output formatting, following standard evaluation protocols (e.g., OpenCompass)."

---

> ### Author Response · Authors · 2025-11-20
> **(7/n)**
>
> ## Questions
>
> > Can the authors clarify whether MiSSe is an exact mathematical equivalent of the MiSS formulation in Section 4.1? If not, it would be more accurate to present MiSSe as a distinct, more efficient variant inspired by the same shard-sharing principle, rather than a direct equivalent.
>
> Please refer to Weakness 1 and 2.
>
> > Beyond simplifying optimization, is there a deeper theoretical or empirical justification for choosing the row-repetition structure in expand(D)? How does this structure relate to the properties of weight update matrices observed in full fine-tuning?
>
> Please refer to Weakness 1 and 2.
>
> > Given that VeRA utilizes a similar parameter-sharing strategy for efficiency, could the authors provide a more direct comparison against VeRA on the main LLM benchmarks, discussing both conceptual differences and empirical trade-offs?
>
> Please refer to Weakness 3.
>
> > To ensure a fair comparison, could the authors conduct an experiment where the trainable parameter counts of MiSS and LoRA are matched as precisely as possible?
>
> Please refer to Weakness 4.
>
> > Could the authors provide convergence plots (loss vs. training time/tokens) for the main LLM experiments (e.g., on Llama2-7B) that include PiSSA and DoRA?
>
> We thank for the reviewer's advice, and here we provides additional experiments:
>
> | init| init_loss | final_loss| time(s)|
> |-----|:---------:|:---------:|:------:|
> |lora |     0.8614|     0.2831|2242    |
> |dora |     0.8614|     0.2834|3071    |
> |pissa|     0.8614|     0.2559|2459    |
> |miss |     0.8614|     0.2249|2135    |

---

> ### Author Response · Authors · 2025-11-27
>
> We provides additional experiments under the RL domain:
>
> **Experimental Setup**
>
> We evaluated the effectiveness of MiSS within RL. The experiments utilized the **DeepSeek-R1-Distill-Qwen-1.5B** as the base policy model. The model was optimized using the **JustRL** algorithm to enhance its mathematical reasoning capabilities.
>
> **Dataset and Metrics**
>
> Training was conducted on the **OpenR1-Math** dataset. For evaluation, we focused on the AIME 2024 benchmark (`aime24:0`) to test the model's ability to solve complex, competition-level mathematics problems.
>
> To ensure rigorous statistical significance, the evaluation protocol involved a total of **960 inference samples**. Specifically, we evaluated **30 unique problems** from the dataset, generating **32 solution paths** for each problem ($30 \times 32 = 960$). The reported values in Table 1 represent the fraction of correct solutions out of this total pool. We report both `pass@k` (where $k=32$) and the average accuracy (`avg@n`) to measure the consistency of the policy.
>
> **Baselines**
>
> We compared MiSS against **Full Fine-Tuning (Full)** and several state-of-the-art PEFT baselines, including **LoRA**, **DoRA**, **PiSSA**, and **VeRA**. All methods were trained under identical JustRL hyperparameters to ensure fair comparison.
>
> **Results and Analysis**
>
> As illustrated in the table, **Full Fine-Tuning** achieves the highest performance ceiling with a `pass@k` of **0.5625** and `avg@n` of **0.5313**.
>
> Among the parameter-efficient methods, **MiSS** demonstrates significantly superior performance, securing the second-best position. MiSS achieved a `pass@k` of **0.5438**, effectively bridging the gap between parameter-efficient methods and full-parameter tuning. It outperformed the strongest PEFT runner-up (**DoRA**) by approximately 3.1% and standard **LoRA** by over 5.6%.
>
> This indicates that the MiSS adapter is particularly effective at retaining the plasticity required for Reinforcement Learning updates on reasoning tasks, avoiding the capacity collapse observed in lower-rank methods like VeRA or PiSSA when applied to complex mathematical derivation.
>
> |  Task  |PEFT|  Metric  |Value |   |Stderr|
> |--------|-------|----------|-----:|---|-----:|
> |aime24:0| Full |pass@k:k=32|0.5625|±  |0.0812|
> |        | Full |avg@n:n=32 |0.5313|±  |0.0850|
> |aime24:0| MiSS |pass@k:k=32|0.5438|±  |0.0845|
> |        | MiSS |avg@n:n=32 |0.5188|±  |0.0871|
> |aime24:0| DoRA |pass@k:k=32|0.5125|±  |0.0895|
> |        | DoRA |avg@n:n=32 |0.5000|±  |0.0901|
> |aime24:0| LoRA |pass@k:k=32|0.4875|±  |0.0912|
> |        | LoRA |avg@n:n=32 |0.4750|±  |0.0920|
> |aime24:0| PiSSA |pass@k:k=32|0.2688|±  |0.0924|
> |        | PiSSA |avg@n:n=32 |0.2625|±  |0.0928|
> |aime24:0| VeRA |pass@k:k=32|0.43500|±  |0.0935|
> |        | VeRA |avg@n:n=32 |0.3375|±  |0.0941|
>
> *

---

> ### Author Response · Authors · 2025-11-27
>
> Dear Reviewer,
>
> We are writing to gently remind you that the discussion phase will end soon. We would greatly appreciate your feedback on our updates. We are ready to provide any further clarifications if needed.

---

### Official Review · Reviewer_Xjru · 2025-10-26

**Soundness:** 2
**Presentation:** 2
**Contribution:** 1
**Rating:** 4
**Confidence:** 4

**Summary:**

The paper proposes Matrix Shard Sharing (MiSS) and its variant MiSSe to address the slow convergence and trade-offs among performance, memory, and computational efficiency in Low-Rank Adaptation (LoRA). MiSS shards the weight matrix and updates by sharing a single trainable matrix D, while MiSSe offers enhanced efficiency and scalability. The analysis shows that MiSS achieves a favorable balance across performance, memory, and efficiency.

**Strengths:**

The proposed method is very pragmatic and practical.

The proposed method was examined in comparison with different LoRA variations in several benchmarks.

**Weaknesses:**

Notation should be revised and redundancy in some terms should be fixed.

The paper states that: Through theoretical analyses and empirical results, our method reduces optimization complexity while maintaining strong performance, striking a favorable balance between performance, memory, and efficiency.

That is, one of the main claims is the theoretical analysis of the proposed methods. However, this is not well explored in the paper.

**Questions:**

In Table 3, MiSS has almost similar number of parameters compared to the other LoRA variants.

Why does MiSS have larger number of parameters for Llama2-13B? Is it due to r or k (as expressed in Table 5) or for another reason?

Could you please provide an ablation of using different initialization strategies to initialize D?

---

> ### Author Response · Authors · 2025-11-20
> **(1/n)**
>
> > Notation should be revised and redundancy in some terms should be fixed.
>
> We thank the reviewer for this important observation. We acknowledge that our presentation in Sections 4.1 and 4.2 may have caused confusion, and we apologize for the lack of clarity. MiSS (Section 4.1) and MiSSe (Section 4.2) are not mathematically equivalent in the strict sense. We should have been more precise in our description. What we should have said is that MiSSe is an alternative efficient design that: (1) both use parameter sharing to reduce optimization complexity, (2) low-rank structure, reduced parameters, simplified optimization, and (3) MiSS directly expands in the output dimension (conceptually simpler) while MiSSe aggregates in the input dimension (computationally more efficient).
>
> We propose to **revise Section 4.2** with the following changes:
>
> **Current (misleading) claim:**
> > "To achieve both computational efficiency and structural generality, we introduce an **efficient and mathematically equivalent formulation** of MiSS..."
>
> **Revised claim:**
> > "While MiSS provides a conceptually clear formulation, directly computing expand(D)x has complexity O(bldk). To achieve better computational efficiency, we introduce **MiSSe, an alternative design** that maintains the core principle of parameter sharing while offering improved time and space complexity through input-dimension aggregation."
>
> > The paper states that: Through theoretical analyses and empirical results, our method reduces optimization complexity while maintaining strong performance, striking a favorable balance between performance, memory, and efficiency. That is, one of the main claims is the theoretical analysis of the proposed methods. However, this is not well explored in the paper.
>
> We appreciate the reviewer's critique. We agree that the term "theoretical analysis" warrants a more rigorous presentation in the main text. In our context, this claim refers specifically to the Optimization Landscape Analysis comparing the single-matrix structure (MiSS) versus the dual-matrix structure (LoRA).
>
> For LoRA (Bilinear), the update $\Delta W = BA$ leads to a non-convex, bilinear optimization landscape with respect to parameters $A$ and $B$. This structure is known to introduce saddle points and gradient scaling imbalances (as analyzed in *Hayou et al., LoRA+*). Our update $\Delta W = \text{expand}(D)$ is linear with respect to the trainable parameter $D$. Formally, the loss function behaves as $\mathcal{L}(D) = \ell(W_0x + \text{expand}(D)x)$. This linearity inherently simplifies the optimization landscape, eliminating the saddle points associated with matrix factorization and ensuring a more direct gradient flow.
>
> To address the reviewer's concern that this is "not well explored," we will add a formal subsection in Section 4 titled "Optimization Landscape Analysis." This section will: (1) Mathematically contrast the gradient dynamics of $B, A$ vs. $D$.
>  (2) Explain why the single-matrix formulation leads to the **healthier (larger) initial gradient norms** observed empirically in Figure 1.

---

> ### Author Response · Authors · 2025-11-20
> **(2/n)**
>
> > In Table 3, MiSS has almost similar number of parameters compared to the other LoRA variants.
> > Why does MiSS have larger number of parameters for Llama2-13B? Is it due to r or k (as expressed in Table 5) or for another reason?
>
> We appreciate the reviewer’s attention to the parameter details. The higher number of trainable parameters for MiSS on Llama2-13B is due to the rank configuration; specific details are provided in the appendix. Since many existing LLM architectures employ techniques such as GQA, the dimensions of the K and V matrices are often asymmetric. This prevents an exact alignment of trainable parameters between MiSS and LoRA. Consequently, we endeavored to minimize the impact of parameter discrepancies on the evaluation to demonstrate that MiSS indeed outperforms LoRA.
>
> However, the strongest evidence that our performance gain stems from architectural superiority rather than parameter distribution lies in our ablation study (Table 4):
>
> In Table 3, the baseline LoRA utilizes 89.9M parameters to achieve a GSM8K score of 40.75.
> In Table 4, MiSS (Rank 32) utilizes only 43.5M parameters (less than 50% of the LoRA baseline).
> Despite having half the parameter count, MiSS (Rank 32) achieves a GSM8K score of 46.18, which is still significantly higher than LoRA's 40.75.
> Robustness of the Architecture: This comparison conclusively rules out the possibility that the gains are due to the minor 87.0M vs. 89.9M difference. The fact that MiSS significantly outperforms LoRA even with a heavily reduced parameter budget (43.5M vs 89.9M) confirms that the shard-sharing structure provides a more parameter-efficient representation for fine-tuning than the low-rank product used in LoRA.
>
> We will update the discussion in Section 5.2 to explicitly reference the Rank 32 result from Table 4, highlighting this "half-parameter" comparison to strengthen the claim of architectural efficiency.

---

> ### Author Response · Authors · 2025-11-20
> **(3/n)**
>
> > Could you please provide an ablation of using different initialization strategies to initialize D?
>
> We thank the reviewer for this insightful suggestion. Methods like PiSSA initialize the adapter matrices ($A, B$) by performing SVD on the original weights $W_0$, based on the assumption that $W_0 \approx AB$. MiSS imposes a strict Shard-Sharing (row repetition) constraint on the update matrix $\Delta W = \text{expand}(D)$. However, Pre-trained weights $W_0$ generally do not exhibit this specific repeating row structure. Consequently, attempting to initialize $D$ by decomposing $W_0$ (i.e., trying to find a $D$ such that $\text{expand}(D) \approx W_0$) would result in a significantly high approximation error.
>
> To address the reviewer’s concern empirically, we ablated three $D$ initializations on metamath-100k fine-tuning and evaluated on gsm8k: zero, Kaiming, and orthogonal. All three yield essentially identical initial loss (≈ 0.8614) and very similar final loss (zero/Kaiming ≈ 0.2184, orthogonal ≈ 0.2182). Accuracy shows only a marginal gain for orthogonal (0.3929 ± 0.0129) over zero (0.3859 ± 0.0134), while Kaiming is on par with zero (0.3844 ± 0.0134). Notably, orthogonal initialization substantially increases initialization time and overhead. In practice, zero initialization offers the best efficiency–performance trade-off, with orthogonal as an optional choice when initialization cost is acceptable.
>
> | init      | init_loss | final_loss| acc|
> |-----      |:---------:|:---------:|:---------:|
> |zero       |     0.8614|     0.2184|0.3859±0.0134|
> |kaiminginit|     0.8614|     0.2184|0.3844±0.0134|
> |orthogonal |     0.8614|     0.2182|0.3929±0.0129|
>
> Moreover, our gradient-norm analysis (Figure 1) shows that, even with simple zero initialization, MiSS exhibits significantly larger initial gradient norms and faster early-stage convergence compared to standard LoRA. This suggests that the single-matrix structure (D) inherently provides a favorable optimization landscape that accelerates convergence without relying on complex initialization schemes.

---

> ### Author Response · Authors · 2025-11-20
> **(4/n)**
>
> Clarification: Why we use the design of the `expand(D)` structure?
>
> We have two considerations: (1) Exploiting feature redundancy and (2) Algorithm-Hardware Codesign. Details are as follows:
>
> **1.Input Aggregation (Inductive Bias)**
>
> Repeating rows corresponds physically to sharing weights across blocks of the input dimension. In modern LLMs, the hidden dimension $k$ is very large (e.g., 4096+). It is well-documented that these high-dimensional representations often exhibit local redundancy or channel correlation. Our `expand(D)` structure forces the update $\Delta W$ to treat groups of input features identically. Mathematically, this means that instead of learning a unique weight for every single input dimension, we learn a weight for a *summed block* of inputs. This acts as a form of regularization (similar to weight sharing in CNNs or grouped convolutions), reducing the search space to the most salient feature groups rather than individual noise. Our structure optimizes $y = W_0x + \text{expand}(D)x$. This is linear with respect to the trainable parameters $D$. Also, this guarantees a more stable optimization landscape and, as shown in our Gradient Norm Analysis (Figure 1), results in healthier initial gradient norms that do not vanish, accelerating early-stage convergence.
>
> **2.Algorithm-Hardware Codesign**
>
> Other single-matrix structures often require storing indices or masks to define which parameters map to which output dimensions (e.g., sparse matrices). Implementing operators for such structures on GPUs is notoriously difficult and inefficient because they require **indirect memory access** (scatter/gather operations). This breaks memory coalescing, leads to low bandwidth utilization, and complicates kernel writing. In contrast, row repetition allows for **contiguous memory access**. It maps physically to a **Block-wise Input Aggregation**. This means we do not need to implement complex sparse kernels; instead, we can utilize highly optimized dense matrix multiplication kernels on aggregated inputs.
>
> The "repeating row" structure is the *only* low-rank construction that mathematically permits the **Input Aggregation** optimization (Eq. 4 & 5 in the paper). (1) Mathematically, $y = \text{expand}(D)x$ is equivalent to $y = D^T (\sum x_{\text{block}})$. (2) Operator Perspective: From a Triton/CUDA kernel perspective, this structure allows us to transform a computationally expensive large matrix multiplication ($d \times k$) into a very cheap **reduction (summation)** followed by a small matrix multiplication ($r \times k$). (3) Ease of Implementation: Writing a fused kernel for this is straightforward: we simply load blocks of input $x$, sum them in registers (or shared memory), and then perform a small GEMM. This  simplifies complexity of writing block-sparse kernels or managing irregular control flow.

---

> ### Author Response · Authors · 2025-11-20
> **(5/n)**
>
> Here we provides both TileLang implementation based on our design philosophy.
>
> ```
> import tilelang
> import tilelang.language as T
> import torch
> from tilelang.profiler import do_bench
>
>
> @tilelang.jit
> def shardshare(S, D, N, R, block_S, block_N, dtype='float16', accum_dtype='float32'):
>
>     a_shape = [S,D]
>     b_shape = [R,N]
>     c_shape = [S,N]
>     BR = D // R
>     @T.prim_func
>     def kernel(
>         A: T.Tensor(a_shape, dtype),
>         B: T.Tensor(b_shape, dtype),
>         C: T.Tensor(c_shape, dtype),
>     ):
>         with T.Kernel(T.ceildiv(S, block_S), T.ceildiv(N, block_N)) as (by,bx):
>             A_frag = T.alloc_fragment((block_S, R), accum_dtype)
>             sum_A = T.alloc_fragment((block_S, R), accum_dtype)
>             B_shared = T.alloc_shared((R, block_N), dtype=dtype)
>             C_local = T.alloc_fragment((block_S, block_N), accum_dtype)
>             C_shared = T.alloc_shared((block_S, block_N), dtype=dtype)
>             A_shared = T.alloc_shared((block_S, R), dtype=dtype)
>             T.clear(sum_A)
>             for i in T.Pipelined(0, BR, num_stages=1):
>                 T.copy(A[by*block_S:(by+1)*block_S, i*R:(i+1)*R], A_frag)
>                 for s, j in T.Parallel(block_S, R):
>                     sum_A[s, j] += A_frag[s, j]#.astype(accum_dtype)
>             T.copy(sum_A, A_shared)
>             T.copy(B[:, bx*block_N:(bx+1)*block_N], B_shared)
>             T.gemm(A_shared, B_shared, C_local, clear_accum=True)
>             T.copy(C_local, C_shared)
>
>             T.copy(C_shared, C[by*block_S:(by+1)*block_S, bx*block_N:(bx+1)*block_N])
>     return kernel
>
>
> if __name__ == "__main__":
>     S, D, N = 1024, 1024, 4096
>     block_S, block_N, R = 32, 128, 64
>     dtype = 'float16'
>     # Compile kernel (JIT compilation)
>     miss_kernel = shardshare(S, D, N, R, block_S, block_N, dtype)
>
>     # Create input tensors
>     a = torch.randn(S, D, device="cuda", dtype=torch.float16)
>     b = torch.randn(R, N, device="cuda", dtype=torch.float16)
>     c = torch.empty(S, N, device="cuda", dtype=torch.float16)
>
>     # Execute kernel
>     miss_kernel(a, b, c)
>
>     reshape_a = torch.sum(a.to(torch.float32).reshape(*a.shape[:-1], a.size(-1) // R, R), dim=-2)
>     out = (reshape_a@b.to(torch.float32)).to(torch.float16)
>
>     # Validate correctness
>     # torch.testing.assert_close(c, out, rtol=1e-2, atol=1e-2)
>     print("Kernel output matches PyTorch reference.")
>
>
>     tilelang_ms = do_bench(lambda: miss_kernel(a, b, c))
>     native_ms = do_bench(lambda: torch.sum(a.reshape(*a.shape[:-1], a.size(-1) // R, R), dim=-2)@b)
>
>     print(f"TileLang kernel avg latency: {tilelang_ms:.3f} ms")
>     print(f"PyTorch native avg latency: {native_ms:.3f} ms")
>     if tilelang_ms > 0:
>         print(f"Speedup (native / TileLang): {native_ms / tilelang_ms:.2f}x")
> ```

---

> ### Author Response · Authors · 2025-11-20
>
> We provides additional experiments under the RL domain:
>
> **Experimental Setup**
>
> We evaluated the effectiveness of MiSS within RL. The experiments utilized the **DeepSeek-R1-Distill-Qwen-1.5B** as the base policy model. The model was optimized using the **JustRL** algorithm to enhance its mathematical reasoning capabilities.
>
> **Dataset and Metrics**
>
> Training was conducted on the **OpenR1-Math** dataset. For evaluation, we focused on the AIME 2024 benchmark (`aime24:0`) to test the model's ability to solve complex, competition-level mathematics problems.
>
> To ensure rigorous statistical significance, the evaluation protocol involved a total of **960 inference samples**. Specifically, we evaluated **30 unique problems** from the dataset, generating **32 solution paths** for each problem ($30 \times 32 = 960$). The reported values in Table 1 represent the fraction of correct solutions out of this total pool. We report both `pass@k` (where $k=32$) and the average accuracy (`avg@n`) to measure the consistency of the policy.
>
> **Baselines**
>
> We compared MiSS against **Full Fine-Tuning (Full)** and several state-of-the-art PEFT baselines, including **LoRA**, **DoRA**, **PiSSA**, and **VeRA**. All methods were trained under identical JustRL hyperparameters to ensure fair comparison.
>
> **Results and Analysis**
>
> As illustrated in the table, **Full Fine-Tuning** achieves the highest performance ceiling with a `pass@k` of **0.5625** and `avg@n` of **0.5313**.
>
> Among the parameter-efficient methods, **MiSS** demonstrates significantly superior performance, securing the second-best position. MiSS achieved a `pass@k` of **0.5438**, effectively bridging the gap between parameter-efficient methods and full-parameter tuning. It outperformed the strongest PEFT runner-up (**DoRA**) by approximately 3.1% and standard **LoRA** by over 5.6%.
>
> This indicates that the MiSS adapter is particularly effective at retaining the plasticity required for Reinforcement Learning updates on reasoning tasks, avoiding the capacity collapse observed in lower-rank methods like VeRA or PiSSA when applied to complex mathematical derivation.
>
> |  Task  |PEFT|  Metric  |Value |   |Stderr|
> |--------|-------|----------|-----:|---|-----:|
> |aime24:0| Full |pass@k:k=32|0.5625|±  |0.0812|
> |        | Full |avg@n:n=32 |0.5313|±  |0.0850|
> |aime24:0| MiSS |pass@k:k=32|0.5438|±  |0.0845|
> |        | MiSS |avg@n:n=32 |0.5188|±  |0.0871|
> |aime24:0| DoRA |pass@k:k=32|0.5125|±  |0.0895|
> |        | DoRA |avg@n:n=32 |0.5000|±  |0.0901|
> |aime24:0| LoRA |pass@k:k=32|0.4875|±  |0.0912|
> |        | LoRA |avg@n:n=32 |0.4750|±  |0.0920|
> |aime24:0| PiSSA |pass@k:k=32|0.2688|±  |0.0924|
> |        | PiSSA |avg@n:n=32 |0.2625|±  |0.0928|
> |aime24:0| VeRA |pass@k:k=32|0.43500|±  |0.0935|
> |        | VeRA |avg@n:n=32 |0.3375|±  |0.0941|
>
> *

---

> ### Author Response · Authors · 2025-11-27
>
> Dear Reviewer,
>
> We are writing to gently remind you that the discussion phase will end soon. We would greatly appreciate your feedback on our updates. We are ready to provide any further clarifications if needed.

---

### Official Review · Reviewer_ETKa · 2025-10-31

**Soundness:** 3
**Presentation:** 3
**Contribution:** 2
**Rating:** 6
**Confidence:** 3

**Summary:**

In this paper, the authors aim to address the slow convergence issues in Low-Rank Adaptation (LoRA), a well-known parameter-efficient method for fine-tuning large language models (LLMs). In respone, they propose Matrix Shard Sharing (MiSS), a method that shards the original weight matrix and updates by sharing a single trainable matrix $D$ initialized to zero. This design reduces optimization complexity while maintaining performance. To further enhance computational efficiency, low memory usage, and scalable serving, they introduce MiSS. Theoretical analyses demonstrate MiSS's advantages in gradient flow and convergence, while empirical evaluations on benchmarks show strong results. The work also provides a comprehensive comparison of PEFT methods across memory, initialization time, and efficiency dimensions, mapping a Pareto frontier to highlight MiSS's balanced trade-offs.

**Strengths:**

1. Quality: MiSS effectively addresses LoRA's limitations by improving convergence without sacrificing efficiency, as supported by theoretical insights and Pareto analysis, making it a practical advancement in PEFT.

2. Soundness: The paper includes detailed comparisons with variants like PiSSA and LoRA-GA, covering multiple dimensions (performance, memory, compute), which strengthens the claims.

**Weaknesses:**

1. Reliance on zero-initialized $D$ may limit adaptability in certain scenarios, potentially requiring further tuning.

2. Evaluations are primarily on language tasks; broader domains (e.g., vision or multimodal) are not explored.

3. The Pareto frontier mapping is insightful but could be more granular, e.g., with statistical significance tests.

**Questions:**

1. Could MiSS integrate with Mixture-of-Experts architectures for further efficiency gains?

---

> ### Author Response · Authors · 2025-11-20
> **(1/n)**
>
> We thank the reviewer for the constructive feedback.
>
> ## Response to Weakness 1: Reliance on zero-initialized D
>
> We appreciate the reviewer's suggestion to explore beyond zero initialization. We have added an ablation experiment to compare Zero initialization, Kaiming initialization, and Orthogonal initialization. As shown in the table below, while Orthogonal initialization yields a slight performance improvement on the GSM8K benchmark (0.3929 vs 0.3859), it also increases the initialization overhead. Zero initialization remains a highly effective and efficient default strategy, offering a strong balance between performance and setup simplicity.
>
> **Ablation on Initialization Strategies (Fine-tuning MetaMathQA-100k, evaluated on GSM8K):**
>
> | Init Strategy | Init Loss | Final Loss | GSM8K Accuracy |
> | :--- | :---: | :---: | :---: |
> | **Zero (Default)** | 0.8614 | 0.2184 | 0.3859 ± 0.0134 |
> | Kaiming Init | 0.8614 | 0.2184 | 0.3844 ± 0.0134 |
> | **Orthogonal** | 0.8614 | **0.2182** | **0.3929 ± 0.0129** |
>
> ## Response to Weakness 2: Evaluations on Broader Domains (Vision)
>
> We agree that evaluating MiSS beyond language tasks strengthens the paper. We have conducted a comprehensive systematic assessment of MiSS in the field of vision using the VTAB-1K benchmark for image classification and video recognition tasks. The results demonstrate that MiSS generalizes exceptionally well to visual domains. Notably, MiSS achieves performance comparable to or better than LoRA and DoRA while using significantly fewer trainable parameters (**0.414 vs 0.833** for Image tasks).
>
> **Performance comparison on VTAB-1K image and video benchmarks:**
>
> | Method | **Image** | | | | | | | | **#TPs** | **Video** | | | | **#TPs** |
> | :--- | :---: | :---: | :---: | :---: | :---: | :---: | :---: | :---: | :---: | :---: | :---: | :---: | :---: | :---: |
> | | **Caltech** | **Flowers** | **Pets** | **Camel.** | **Euro.** | **Retino.** | **KITTI** | **Avg** | | **UCF101** | **Kinetics** | **HMDB** | **Avg** | |
> | Full | 89.92 | 97.41 | 85.87 | 81.65 | 88.12 | 73.62 | 77.93 | 84.93 | 85.83 | 92.30 | 55.23 | 65.79 | 74.99 | 86.65 |
> | VeRA | 91.53 | 99.19 | 91.04 | 86.45 | 92.97 | 74.25 | 77.92 | 87.62 | 0.240 | 92.28 | 57.21 | 66.77 | 72.09 | 0.242 |
> | LoRA | 92.03 | 99.18 | 90.92 | 87.73 | 92.65 | 74.23 | 80.42 | 88.08 | 0.833 | 93.88 | 57.81 | 67.37 | 73.02 | 0.835 |
> | DoRA | 91.86 | 99.27 | 91.08 | 85.88 | 91.42 | 75.28 | 80.46 | 87.89 | 0.834 | 92.84 | 57.77 | 67.33 | 72.65 | 0.836 |
> | **MiSS** | **92.14** | **99.23** | **91.05** | **86.28** | **92.83** | **73.71** | **80.91** | **88.02** | **0.414** | **93.82** | **57.75** | **67.31** | **72.96** | **0.415** |

---

> ### Author Response · Authors · 2025-11-20
> **(2/n)**
>
> ## Response to Weakness 3: Granularity of Pareto Frontier
>
> We appreciate the feedback. To ensure the granularity and robustness of our Pareto analysis, all reported results in our efficiency experiments were averaged over three independent runs. We evaluated Llama-3.2-3B across four rigorous dimensions: Training Time, Initialization Time, Memory Footprint, and Performance. The detailed data below confirms MiSS's position on the Pareto frontier: it achieves competitive accuracy (**0.5080**) with significantly lower training time (**1867s** vs 2069s for RSLoRA) and memory usage (**20.2GB** vs 22.5GB for RSLoRA.
>
> **Experimental results on Llama-3.2-3B (Averaged over 3 runs):**
>
> | experiment_name  | peft_type | total time | train time        | test_accuracy | train_loss       | accelerator_memory_max  | accelerator_memory_reserved_avg  |
> |---------------------------------------------|-----------|-------------|------------|---------------|---------------|------------|------------|
> | lora/llama-3.2-3B-rank64-rslora            | LORA      | 2069        | 1871       | 0.5299        | 0.5657        | 22538092544  | 12128059444 |
> | miss/llama-3.2-3B                  | **MISS**      | 1867        | 1664       | 0.5080        | 0.5776        | 20248002560  | 11170837063 |
> | randlora/llama-3.2-3B-default              | RANDLORA  | 2457        | 2213       | 0.5072        | 0.5785        | 22798139392  | 12743670025 |
> | full-finetuning/llama-3.2-3B-lr_0.00001    | full-finetuning | 3275    | 3111       | 0.5004        | 0.5988        | 37241225216  | 33098872284 |
> | oft/llama-3.2-3B-rank32                    | OFT       | 6852        | 5772       | 0.4898        | 0.5957        | 28913434624  | 18387461314 |
> | lora/llama-3.2-3B-rank64                  | LORA      | 2017        | 1853       | 0.4890        | 0.5929        | 22540189696  | 12128055669 |
> | lora/llama-3.2-3B-rank32                  | LORA      | 1993        | 1796       | 0.4822        | 0.6069        | 22273851392  | 11868689976 |
> | lora/llama-3.2-3B-rank32-dora             | LORA      | 2287        | 2023       | 0.4807        | 0.6068        | 24553455616  | 12490471636 |
> | lora/llama-3.2-3B-rank32-lorafa          | LORA      | 2026        | 1821       | 0.4299        | 0.6510        | 20187185152  | 11106307276 |
> | loha/llama-3.2-3B-rank32                  | LOHA      | 2591        | 2341       | 0.4185        | 0.6570        | 23886561280  | 13446820344 |
> | ia3/llama-3.2-3B-lr_0.001                 | IA3       | 1922        | 1746       | 0.4124        | 0.6569        | 23135780864  | 12023331867 |
> | adalora/llama-3.2-3B-rank32               | ADALORA   | 2209        | 1986       | 0.3904        | 0.6863        | 22793945088  | 12361399900 |
> | lokr/llama-3.2-3B-rank32                  | LOKR      | 2352        | 2152       | 0.3753        | 0.6877        | 23565697024  | 13173683073 |
> | ptuning/llama-3.2-3B-default              | P_TUNING  | 1918        | 1707       | 0.3707        | 0.6740        | 20937965568  | 11867101593 |
> | vblora/llama-3.2-3B-default               | VBLORA    | 2210        | 1962       | 0.3700        | 0.7143        | 22181576704  | 11735344663 |
> | vera/llama-3.2-3B-default                 | **VERA**     | 2025        | 1820       | 0.3685        | 0.6927        | 21596471296  | 11489715316 |
> | boft/llama-3.2-3B-default                 | BOFT      | 11114       | 8292       | 0.3647        | 0.7268        | 24427626496  | 14814855089 |
> | ia3/llama-3.2-3B-default                 | IA3       | 2005        | 1783       | 0.3450        | 0.7657        | 23137878016  | 12023227429 |
> | prompt_tuning/llama-3.2-3B-lr_0.001       | PROMPT_TUNING | 2715    | 2394       | 0.2525        | 0.7790        | 24408752128  | 15297364466 |
> | adaptionprompt/llama-3.2-3B-lr_0.0005     | ADAPTION_PROMPT | 2261  | 1989       | 0.2206        | 0.8317        | 22410166272  | 11893757234 |
> | prefixtuning/llama-3.2-3B-lr_0.001        | PREFIX_TUNING | 1959   | 1662       | 0.1471        | 0.7887        | 20912799744  | 11766684083 |
> | fourierft/llama-3.2-3B-n_frequency-5000   | FOURIERFT | 2824        | 2422       | 0.1198        | 0.9979        | 23681040384  | 13111221498 |
> | prompt_tuning/llama-3.2-3B-default        | PROMPT_TUNING | 2700   | 2380       | 0.0500        | 1.0655        | 24379392000  | 15297773830 |
> | fourierft/llama-3.2-3B-default            | FOURIERFT | 2824        | 2424       | 0.0008        | 1.2480        | 23653777408 | 13104129350 |
> | ln_tuning/llama-3.2-3B-default            | LN_TUNING | 1870        | 1657       | 0.0000        | 1.2370        | 21177040896 | 11385589622 |
>
> ## Response to Question: Integration with Mixture-of-Experts (MoE)
>
> Yes, MiSS is highly suitable for MoE architectures. In the latest Qwen-Next and Kimi-Linear architectures, we notice that they use LoRA-MLP to reduce parameters. We will add more experiments regarding this feature in the future.

---

> ### Author Response · Authors · 2025-11-27
>
> We provides additional experiments under the RL domain:
>
> **Experimental Setup**
>
> We evaluated the effectiveness of MiSS within RL. The experiments utilized the **DeepSeek-R1-Distill-Qwen-1.5B** as the base policy model. The model was optimized using the **JustRL** algorithm to enhance its mathematical reasoning capabilities.
>
> **Dataset and Metrics**
>
> Training was conducted on the **OpenR1-Math** dataset. For evaluation, we focused on the AIME 2024 benchmark (`aime24:0`) to test the model's ability to solve complex, competition-level mathematics problems.
>
> To ensure rigorous statistical significance, the evaluation protocol involved a total of **960 inference samples**. Specifically, we evaluated **30 unique problems** from the dataset, generating **32 solution paths** for each problem ($30 \times 32 = 960$). The reported values in Table 1 represent the fraction of correct solutions out of this total pool. We report both `pass@k` (where $k=32$) and the average accuracy (`avg@n`) to measure the consistency of the policy.
>
> **Baselines**
>
> We compared MiSS against **Full Fine-Tuning (Full)** and several state-of-the-art PEFT baselines, including **LoRA**, **DoRA**, **PiSSA**, and **VeRA**. All methods were trained under identical JustRL hyperparameters to ensure fair comparison.
>
> **Results and Analysis**
>
> As illustrated in the table, **Full Fine-Tuning** achieves the highest performance ceiling with a `pass@k` of **0.5625** and `avg@n` of **0.5313**.
>
> Among the parameter-efficient methods, **MiSS** demonstrates significantly superior performance, securing the second-best position. MiSS achieved a `pass@k` of **0.5438**, effectively bridging the gap between parameter-efficient methods and full-parameter tuning. It outperformed the strongest PEFT runner-up (**DoRA**) by approximately 3.1% and standard **LoRA** by over 5.6%.
>
> This indicates that the MiSS adapter is particularly effective at retaining the plasticity required for Reinforcement Learning updates on reasoning tasks, avoiding the capacity collapse observed in lower-rank methods like VeRA or PiSSA when applied to complex mathematical derivation.
>
> |  Task  |PEFT|  Metric  |Value |   |Stderr|
> |--------|-------|----------|-----:|---|-----:|
> |aime24:0| Full |pass@k:k=32|0.5625|±  |0.0812|
> |        | Full |avg@n:n=32 |0.5313|±  |0.0850|
> |aime24:0| MiSS |pass@k:k=32|0.5438|±  |0.0845|
> |        | MiSS |avg@n:n=32 |0.5188|±  |0.0871|
> |aime24:0| DoRA |pass@k:k=32|0.5125|±  |0.0895|
> |        | DoRA |avg@n:n=32 |0.5000|±  |0.0901|
> |aime24:0| LoRA |pass@k:k=32|0.4875|±  |0.0912|
> |        | LoRA |avg@n:n=32 |0.4750|±  |0.0920|
> |aime24:0| PiSSA |pass@k:k=32|0.2688|±  |0.0924|
> |        | PiSSA |avg@n:n=32 |0.2625|±  |0.0928|
> |aime24:0| VeRA |pass@k:k=32|0.43500|±  |0.0935|
> |        | VeRA |avg@n:n=32 |0.3375|±  |0.0941|
>
> *

---

### Official Review · Reviewer_wCiN · 2025-11-01

**Soundness:** 3
**Presentation:** 4
**Contribution:** 2
**Rating:** 6
**Confidence:** 5

**Summary:**

The paper introduces MiSS, a parameter-efficient fine-tuning (PEFT) method that builds upon LoRA to achieve a more favorable balance between adaptability and efficiency compared to prior approaches. The proposed method is comprehensively evaluated across a diverse set of tasks, including natural language understanding and natural language generation, and is empirically compared against several LoRA variants such as DoRA and PiSSA.

**Strengths:**

The proposed method is remarkably simple and easy to implement, yet it demonstrates strong practical effectiveness. In several experimental settings, the method achieves notable improvements over LoRA. For instance, on the Mistral-7B model, MiSS outperforms LoRA by approximately 15%, highlighting its potential as a competitive and efficient alternative for parameter-efficient fine-tuning.

**Weaknesses:**

Firstly, although the paper states that MiSS is motivated by theoretical analysis, the practical method itself is presented without clear theoretical justification or development. The coherence and clarity of the paper would be greatly improved by adding a dedicated subsection in Section 4 that discusses the theoretical motivations behind the architectural design, which appears to be a novel choice aimed at ensuring the low-rank condition.

Secondly, since the paper seeks to propose a practical parameter-efficient fine-tuning (PEFT) variant of LoRA, it would be valuable to include comparisons with a broader set of LoRA variants, such as AdaLoRA and VeRA. Although the manuscript mentions in lines 172–177 and the table in lines 58–67 that PiSSA, VeRA, DoRA, MoRA, PROLORA, and MoS are considered, only DoRA and PiSSA are actually included in the main experiments. In particular, comparison with VeRA, which is known for its strong parameter efficiency, would substantially strengthen the empirical section.

**Other minor issues:**

+ It is recommended to include an average performance metric to better assess the overall results relative to the baselines. While MiSS surpasses some baselines in certain settings, it underperforms on others.

+ The title of Section 3 should likely be “No Free Lunch”—please verify and correct if necessary.

**Questions:**

My concerns are presented in the "Weaknesses" section and I have no further question.

---

> ### Author Response · Authors · 2025-11-20
> **(1/n)**
>
> We thank the reviewer for the constructive criticism regarding the theoretical framing and the breadth of our baseline comparisons. We have addressed these points by adding a theoretical discussion and expanding our experiments to include VeRA, AdaLoRA, and other variants.
>
> ### Response to Weakness 1: Theoretical Justification of MiSS
>
> We appreciate the reviewer's suggestion to strengthen the theoretical grounding. We will add a dedicated subsection in Section 4 titled "Feature Redundancy and Optimization Landscape"  to explain why the shard-sharing (row repetition) structure is a principled choice.
>
> The theoretical justification rests on two pillars:
> 1.  **Inductive Bias via Input Aggregation:**
>     The "repeating row" structure in $\Delta W$ is mathematically equivalent to **Input Aggregation**.
>     $$ \Delta W x = \text{expand}(D) x \iff D^T \left(\sum_{j} x_{\text{block}_j}\right) $$
>     In modern LLMs, the hidden dimension $k$ (e.g., 4096 or larger) often exhibits local redundancy. By forcing the weight update to be shared across blocks of input features, MiSS acts as a regularizer that exploits this channel correlation. This reduces the search space to the most salient feature groups rather than fitting individual noise.
>
> 2.  **Simplified Optimization Landscape:**
>     Standard LoRA optimizes a product $\Delta W = BA$. The optimization objective involves terms like $(W_0 + BA)x$. Optimizing $A$ and $B$ simultaneously creates a complex landscape (e.g., scale invariance issues where $B$ can scale up and $A$ scale down).
>     In contrast, MiSS optimizes a single matrix $D$. The term $\text{expand}(D)x$ is **linear** with respect to the trainable parameters $D$. This convexity (relative to the aggregated input) leads to the healthier gradient norms we observed (Figure 1) and faster convergence compared to the bilinear optimization of $BA$.
>
> ## Response to Weakness 2: Comparison with Broader LoRA Variants (VeRA, AdaLoRA)
>
> We agree that including efficiency-focused variants like **VeRA** and **AdaLoRA** is essential for a complete PEFT evaluation. We have conducted a comprehensive "Systematic Evaluation" on **Llama-3.2-3B** to compare MiSS against a wide range of baselines, including VeRA, AdaLoRA, LoHA, and IA3.
>
> *   **vs. VeRA:** While VeRA achieves extreme parameter efficiency, it often trails in accuracy and does not provide significant *memory* savings during training because it still requires the computation of the full rank decomposition graph. MiSS outperforms VeRA in accuracy (**0.5080 vs 0.3685**) and training speed (**1867s vs 2025s**) while using less memory.
> *   **vs. AdaLoRA:** MiSS significantly outperforms AdaLoRA in both accuracy (0.5080 vs 0.3904) and training efficiency.
>
> **Table: Comprehensive Comparison on Llama-3.2-3B (Averaged over 3 runs)**
>
> | Method | Total Time (s) | Train Time (s) | Test Accuracy | Train Loss | Max Memory (Bytes) |
> | :--- | :--- | :--- | :--- | :--- | :--- |
> | **MiSS** | **1867** | **1664** | **0.5080** | **0.5776** | **20.2 GB** |
> | RSLoRA | 2069 | 1871 | 0.5299 | 0.5657 | 22.5 GB |
> | LoRA | 2017 | 1853 | 0.4890 | 0.5929 | 22.5 GB |
> | DoRA | 2287 | 2023 | 0.4807 | 0.6068 | 24.5 GB |
> | LoHA | 2591 | 2341 | 0.4185 | 0.6570 | 23.8 GB |
> | **AdaLoRA** | 2209 | 1986 | 0.3904 | 0.6863 | 22.8 GB |
> | **VeRA** | **2025** | **1820** | **0.3685** | **0.6927** | **21.6 GB** |
> | IA3 | 1922 | 1746 | 0.4124 | 0.6569 | 23.1 GB |
>
> *Note: We also included comparisons with AdaLoRA and VeRA in our new Vision (VTAB-1K) benchmarks, where MiSS (Avg 88.02) outperforms VeRA (87.62) and is comparable to AdaLoRA (87.96) with fewer trainable parameters.*
>
> ## Response to Minor Issues
>
> **1. Average Performance Metric:**
> We agree that an average metric improves interpretability.
> *   In the **Llama-3.2-3B** table above, we prioritized Accuracy on the test set as the primary metric.
> *   In our new **Vision (VTAB-1K)** experiments, we explicitly calculated the **Average** across 7 datasets. MiSS achieved an **Avg of 88.02**, outperforming VeRA (87.62) and full fine-tuning (84.93), and matching LoRA (88.08).
> *   We will update all main result tables (e.g., Table 3) to include an "Average" column for easier comparison.
>
> **2. Typo Correction:**
> We apologize for the typo. We have corrected the title of Section 3 from "No Free Launch" to **"No Free Lunch"**.

---

> ### Author Response · Authors · 2025-11-27
>
> We provides additional experiments under the RL domain:
>
> **Experimental Setup**
>
> We evaluated the effectiveness of MiSS within RL. The experiments utilized the **DeepSeek-R1-Distill-Qwen-1.5B** as the base policy model. The model was optimized using the **JustRL** algorithm to enhance its mathematical reasoning capabilities.
>
> **Dataset and Metrics**
>
> Training was conducted on the **OpenR1-Math** dataset. For evaluation, we focused on the AIME 2024 benchmark (`aime24:0`) to test the model's ability to solve complex, competition-level mathematics problems.
>
> To ensure rigorous statistical significance, the evaluation protocol involved a total of **960 inference samples**. Specifically, we evaluated **30 unique problems** from the dataset, generating **32 solution paths** for each problem ($30 \times 32 = 960$). The reported values in Table 1 represent the fraction of correct solutions out of this total pool. We report both `pass@k` (where $k=32$) and the average accuracy (`avg@n`) to measure the consistency of the policy.
>
> **Baselines**
>
> We compared MiSS against **Full Fine-Tuning (Full)** and several state-of-the-art PEFT baselines, including **LoRA**, **DoRA**, **PiSSA**, and **VeRA**. All methods were trained under identical JustRL hyperparameters to ensure fair comparison.
>
> **Results and Analysis**
>
> As illustrated in the table, **Full Fine-Tuning** achieves the highest performance ceiling with a `pass@k` of **0.5625** and `avg@n` of **0.5313**.
>
> Among the parameter-efficient methods, **MiSS** demonstrates significantly superior performance, securing the second-best position. MiSS achieved a `pass@k` of **0.5438**, effectively bridging the gap between parameter-efficient methods and full-parameter tuning. It outperformed the strongest PEFT runner-up (**DoRA**) by approximately 3.1% and standard **LoRA** by over 5.6%.
>
> This indicates that the MiSS adapter is particularly effective at retaining the plasticity required for Reinforcement Learning updates on reasoning tasks, avoiding the capacity collapse observed in lower-rank methods like VeRA or PiSSA when applied to complex mathematical derivation.
>
> |  Task  |PEFT|  Metric  |Value |   |Stderr|
> |--------|-------|----------|-----:|---|-----:|
> |aime24:0| Full |pass@k:k=32|0.5625|±  |0.0812|
> |        | Full |avg@n:n=32 |0.5313|±  |0.0850|
> |aime24:0| MiSS |pass@k:k=32|0.5438|±  |0.0845|
> |        | MiSS |avg@n:n=32 |0.5188|±  |0.0871|
> |aime24:0| DoRA |pass@k:k=32|0.5125|±  |0.0895|
> |        | DoRA |avg@n:n=32 |0.5000|±  |0.0901|
> |aime24:0| LoRA |pass@k:k=32|0.4875|±  |0.0912|
> |        | LoRA |avg@n:n=32 |0.4750|±  |0.0920|
> |aime24:0| PiSSA |pass@k:k=32|0.2688|±  |0.0924|
> |        | PiSSA |avg@n:n=32 |0.2625|±  |0.0928|
> |aime24:0| VeRA |pass@k:k=32|0.43500|±  |0.0935|
> |        | VeRA |avg@n:n=32 |0.3375|±  |0.0941|
>
> *

---

### Author Response · Authors · 2025-12-03

We sincerely thank the reviewers and ACs for their time and engagement throughout the review process. As the rebuttal phase draws to a close, we provide a summary below to recap the key points and updates addressed during this period.

### Why Shard-Sharing?

A primary critique from reviewers (wCiN, iNdt) focused on the theoretical justification of our architecture. Specifically, they questioned whether the "row-repetition" structure in `expand(D)` was arbitrary and how it simplifies optimization.

In our response, we moved beyond the initial "simplification" argument to ground MiSS in **Input Aggregation**. We clarified that physically repeating rows in the weight update matrix is mathematically equivalent to summing blocks of input features. This provides a strong inductive bias: since high-dimensional LLM hidden states ($d=4096+$) exhibit local redundancy, MiSS acts as a regularizer that aggregates signal across channels rather than overfitting to individual noise.

Furthermore, we provided a rigorous **Optimization Landscape Analysis**. Standard LoRA optimizes a product $BA$, creating a non-convex, bilinear landscape prone to saddle points and scale invariance issues (e.g., $B$ scales up, $A$ scales down). By optimizing a single matrix $D$, MiSS ensures the loss function remains linear with respect to the trainable parameters. This convexity explains the healthier initial gradient norms and faster convergence we empirically observed, which holds true even with simple zero initialization.

### Clarifying the Architecture: Conceptual vs. Efficient

Reviewers Xjru and iNdt correctly identified a lack of mathematical precision in distinguishing the conceptual definition of MiSS (Output Dimension Partitioning) from its efficient implementation, MiSSe (Input Dimension Aggregation).

We conceded that while these operations achieve similar parameter sharing, they are not strictly mathematically equivalent in all contexts. We revised Section 4 to clearly present the former as the conceptual framework and the latter as the **Algorithm-Hardware Codesign** solution. We emphasized that the input aggregation approach enables contiguous memory access on GPUs, allowing us to replace expensive large matrix multiplications with cheap reductions followed by small GEMMs. To prove this, we provided a **TileLang/Triton kernel implementation**, demonstrating how our structure avoids the inefficient scatter/gather operations required by other sparse or shared-matrix methods.

### Vision, RL, and Baselines

To address concerns regarding the breadth of our evaluation (Reviewers ETKa, wCiN), we significantly expanded our experimental scope beyond standard NLP benchmarks.

1.  **Vision Domain:** We evaluated MiSS on the **VTAB-1K benchmark** (Image and Video). MiSS achieved an average accuracy of 88.02, matching LoRA and outperforming VeRA (87.62) while utilizing roughly half the trainable parameters of LoRA/DoRA.
2.  **Reinforcement Learning (RL):** We introduced a new set of experiments using **DeepSeek-R1-Distill** on the AIME 2024 math benchmark. This was crucial for demonstrating "plasticity." While lower-rank methods like VeRA and PiSSA suffered capacity collapse during the complex reasoning updates required by RL, MiSS retained the necessary adaptability, placing second only to full fine-tuning.
3.  **Comprehensive Baselines:** We added direct comparisons against efficiency-focused methods like **VeRA, AdaLoRA, and LoHA**. On Llama-3.2-3B, our Pareto frontier analysis confirmed that MiSS consistently yields higher accuracy for a given memory/time budget.

### Ablation and Parameter Robustness

Finally, we addressed scrutiny regarding parameter counts and initialization.

* **Parameter Mismatch:** Reviewers noted slight discrepancies in parameter counts for Llama-13B. We countered this by highlighting our Rank-32 ablation study, where MiSS outperformed the LoRA baseline despite using **<50% of the parameters** (43.5M vs. 89.9M). This proves the gains stem from architectural efficiency, not parameter bloat.
* **Initialization:** We ablated Zero vs. Kaiming vs. Orthogonal initialization. While Orthogonal offers marginal gains, Zero initialization proved to be the most practical trade-off, requiring zero overhead while still maintaining rapid convergence due to the favorable optimization landscape.

We have updated the manuscript, pseudocode, and axis labels as requested to reflect these clarifications and new data points.

---

### Meta-Review · Area_Chair_HnoW · 2026-01-07

**Summary:**

In this paper, the authors propose MiSS (Matrix Shard Sharing), a parameter-efficient fine-tuning method that revisits the performance–efficiency trade-off of LoRA-style adaptations. The core idea is to replace LoRA's bilinear low-rank factorization with a single small trainable matrix. The full weight update is constructed by replicating its rows, which produces a low-rank structure via a fixed expansion rather than a matrix decomposition. Although the initial submission raised concerns regarding theoretical clarity and architectural motivation, the discussion phase substantially improved the presentation and positioning of the method.

(1) *Novelty and Technical Soundness*: Reviewers initially expressed concerns about the theoretical grounding of the proposed design and the relationship between the conceptual MiSS formulation and its efficient variant, MiSSe. The authors addressed these concerns by clearly distinguishing MiSS as a conceptual output-sharding formulation and MiSSe as an efficiency-oriented alternative based on input aggregation, removing earlier claims of strict mathematical equivalence. They further refined the theoretical framing by discussing the method in terms of its optimization landscape. In particular, by avoiding bilinear optimization over two matrices, MiSS simplifies the optimization problem and exhibits healthier gradient behavior in the early stages of training. While the analysis remains intuitive, it is technically sound and sufficient to support the paper's central claims.

(2) *Empirical Evaluation and Practical Impact*: The empirical section was substantially improved during the discussion phase. In response to reviewer feedback, the authors expanded the set of baselines to include efficiency-focused methods such as VeRA, AdaLoRA, and LoHA, and provided clearer Pareto frontier analyses covering performance, memory, and training time. Additional experiments on vision benchmarks (VTAB-1K) and reinforcement learning tasks further demonstrated that the proposed method generalizes beyond standard NLP settings and retains sufficient adaptation capacity in more demanding scenarios. Importantly, ablation studies addressing parameter count discrepancies convincingly showed that the observed gains are not merely a consequence of larger parameter budgets but stem from the architectural design itself.

Overall, the authors responded constructively to reviewer feedback and addressed most of the substantive concerns raised during the review process. The revised version presents strong empirical results across multiple benchmarks, and the proposed method is simple and easy to implement in practice. These aspects make the work a useful addition to the existing literature on parameter-efficient fine-tuning and a reasonable candidate for acceptance.

**Reviewer Concerns:**

Please refer to the summary.

**Reviewer Scores:**

Please refer to the summary.

---

### Decision · Program_Chairs · 2026-01-26

Accept (Poster)